# Safeguard User Privacy in LLM Cloud Services

## Abstract

Large language models (LLMs) have witnessed substantial growth in recent years. To leverage convenient LLM cloud services, users are inevitable to upload their prompts. Further, for tasks such as translation, reading comprehension, and summarization, related files or contexts are inherently required to be uploaded, whether they contain user privacy or not. Despite the rapid advancement of LLM capability, there has been a scarcity of research focusing on preserving user privacy during inference. To this end, this paper conducts a comprehensive study in this domain. Firstly, we demonstrate that (1) the embedding space of tokens is remarkably sparse, and (2) LLMs primarily function in the orthogonal subspace of embedding space, these two factors making privacy extremely vulnerable. Then, we analyze the structural characteristics of LLMs and design a distributed privacy-preserving inference paradigm which can effectively resist privacy attacks. Finally, we conduct a comprehensive evaluation of the defended models on mainstream tasks and find that low-bit quantization techniques can be well combined with our inference paradigm, achieving a balance between privacy, utility, and runtime memory efficiency.

## 1 Introduction

In recent years, LLMs have achieved substantial advancements, enabling machines to undertake various tasks through instructions in natural language form (Radford et al., 2019; Touvron et al., 2023). Despite the simple chatting uses, existing work has shown that supplying some extra prompts is beneficial for fully unleashing the potential of LLMs (e.g., in-context learning) (Brown et al., 2020). In particular, for some context-based tasks such as translation, reading comprehension and summary extraction, users inherently need to supply relevant information (e.g., by using RAG (Lewis et al., 2020)) from their personal databases as part of the prompt to the LLM APIs. A typical example is the integration of the latest GPTs (GPT-4o, GPT-4-turbo) (Achiam et al., 2023) in Microsoft Word and Excel, which are two widely used software across the globe. Users can simply select a portion of text or data and GPT can automatically treat them as contexts for various effective operations such as translation, continuation, or computation. This undoubtedly offers significant convenience to our daily work routines. However, when the relevant text or data involves industry, business or personal privacy—which we believe to be quite common in Word and Excel documents—the use of LLM cloud services as an auxiliary tool poses a risk of privacy breaches.

It appears that we are trapped in a dilemma: to benefit from the convenient cloud services of LLMs, we must compromise on privacy. A straightforward solution is to deploy LLMs on users' personal devices (Lin et al., 2024). However, not all LLM service providers are willing to do this. Further, users may also lack the hardware resources necessary to deploy and run LLMs locally. There is also another potentially viable method, i.e., differential privacy (DP) (Dwork, 2006), which ensures privacy by carefully designed perturbations and has shown promise in several LLM training and fine-tuning tasks (Li et al., 2023; Liu et al., 2024). However, Hu et al. (2024) argue that even a privacy budget in DP that was originally sufficient for protecting privacy can lead to complete privacy leakage when adversaries enhance the attacks, thus rendering the original privacy guarantees limiting.

In the inference phase, perturbation-based methods typically mitigate the leakage of privacy by perturbing or replacing the token embeddings (Zhang et al., 2024b; Edemacu & Wu, 2024). Nevertheless, we hold a slightly negative outlook towards the direct use of these methods in LLMs' inference phase. In this paper, through a comprehensive analysis, we will demonstrate that only substantial perturbations can effectively prevent adversaries from recovering the original data, while such perturbations can lead to a significant decline in model utility on challenging tasks (e.g., math, and we believe there

are scenarios where users upload files or data and let the LLMs perform some statistics or calculations on the information contained within). In our perspective, a practical privacy-preserving method should meet the following criteria: (1) it is effective in resisting advanced attacks; (2) it minimally impacts the utility of LLMs; (3) it is easy to implement. Through an in-depth analysis of the structural characteristics of mainstream open-source LLMs, this paper proposes a novel privacy-preserving method that simultaneously fulfills these three requirements to a certain extent.

**Our Contribution.** We propose a privacy-preserving inference paradigm for LLM cloud services and test its performance across various tasks including general benchmarks, common-sense reasoning, mathematics, coding, and reading comprehension, with few-shot (Brown et al., 2020), zero-shot or chain-of-thought (CoT) (Wei et al., 2022) settings. Our contributions can be summarized as follows:

- We find that the embedding space of tokens is incredibly sparse, with the embeddings of different tokens maintaining a considerable "distance" from one another. In addition, LLMs seldom alter the projection of hidden states within the embedding space in the shallow layers. These two factors are the primary causes for the difficulty in safeguarding user privacy, also for this reason, we demonstrate that simply perturbing the embeddings is insufficient to effectively defend against privacy leakage attacks.

- Building upon the aforementioned two findings, and in conjunction with our analysis on the model structure, we propose a distributed privacy-preserving inference paradigm. Our method enhances the difficulty of attacks by employing a direction-maintained stochastic scaling transformation of the hidden states along with an adaptive compensation mechanism, thereby ensuring privacy without compromising utility.

- We validate the effectiveness and practicality of the proposed method through extensive experiments. Additionally, we find that the proposed defense method exhibits strong compatibility with low-bit quantization techniques, without necessitating any post-quantization calibrations. Our quantized defense strategy can further provide a balanced guarantee for privacy, model utility, and memory efficiency.

## 2 RELATED WORK

**Privacy in LLMs.** Privacy-reconstruction attacks and defenses for AI models has been extensive studied in recent years (Wen et al., 2022; Ye et al., 2023), with the majority of these efforts focusing on traditional models. In the domain of LLMs, related research is still in its infancy. For protecting privacy in the training or fine-tuning phase of LLMs, in addition to the widely studied federated learning paradigm (Tian et al., 2022; Zhao et al., 2023), methods based on DP have also gained attention. For instance, (Yue et al., 2022; Liu et al., 2024) propose to perturb the embeddings of the original training text and then fine-tune the LLM either directly or using PEFT methods. As this paper focuses on the inference, detailed introduction to these methods will not be provided here.

In the inference phase of LLMs, privacy-preserving for the Personally Identifiable Information (PII) has been a subject of study. On the attack side, Kim et al. (2024) and Carlini et al. (2021) have carefully designed the prompts and successfully obtained the PII information in training data of LLMs. In terms of defense, Kan et al. (2023) and Chen et al. (2023) have proposed sanitization-based methods to filter sensitive PII, thereby protecting user privacy. Moreover, other research, which aims to protect all prompts, rather than just PII, has also emerged in recent years. For example, DP-based methods (Zhang et al., 2024b) realize the protection of prompts by perturbing the embeddings or mapping tokens to the nearby tokens. Specifically, Tong et al. (2023) and Mai et al. (2024) perturb the embeddings of prompts before inputting them into the LLM. After the LLM returns a noisy output, they use a local denoising module to correct the LLM's output. In addition to the DP-based methods, Zhang et al. (2024a) have proposed a novel interaction protocol where users send multiple tokens (including real tokens) to the server each time to confuse the server and protect privacy. Differently, Tang et al. (2024) treat the examples for in-context learning as privacy and assume the server as the victim, proposing a method to protect server's examples. *Unfortunately, almost all of the studies (most are preprints) mentioned above have not been tested on mainstream LLM benchmarks (e.g., reasoning, math, code, et al.) comprehensively, so their practicality remains to be further explored.*

**Distributed paradigm in LLMs.** The distributed paradigm here refers to the serial training or inference of LLMs by multiple parties (akin to split learning (Gupta & Raskar, 2018; Kang et al., 2023)). In relevant studies, Zhou et al. (2023) have proposed a user-server collaborative training scheme, which aims to densify the representations of similar words within the user's dataset, thereby increasing the difficulty of privacy attacks. In addition, Wang et al. (2023) and Gao & Zhang (2024) have employed LoRA (Hu et al., 2021) to fine-tune models in a distributed way, aiming to obtain personalized LLMs without compromising privacy. While Borzunov et al. (2024) focus on the scenario of limited hardware resources at the user-side, and have proposed a protocol to invoke online idle GPUs to realize the distributed fine-tuning or inference serially. These works have all demonstrated the feasibility of distributed inference, which can serve as the foundation for our study.

## 3 METHODOLOGY

### 3.1 THREAT MODEL

For the threat model, we assume the victims are users of LLM cloud services who want to obtain the desired feedback by accessing the provided APIs with prompts. Concurrently, we consider the adversary to be a potentially malicious service provider. The adversary aims to obtain users' original data through carefully designed attack strategies when privacy-preserving methods are adopted by the users. Since the most commonly employed defense mechanism currently involves randomly perturbing the token embeddings or hidden states (Edemacu & Wu, 2024), we assume that adversaries are capable of adopting advanced attack strategies against perturbation-based defense mechanisms. The overview of the threat model is shown in Fig. 1 (a).

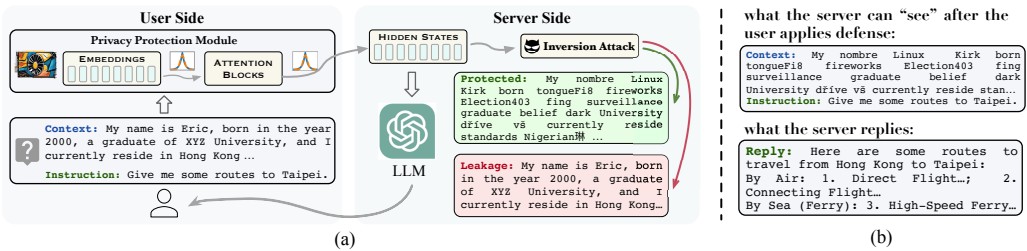

Figure 1: Overview of the threat model, where (a) users aim to obtain LLMs services while safeguarding their privacy, whereas adversaries seek to obtain user privacy during the provision of services; (b) shows the ideal scenario where the server can respond accurately without being able to see the data.

In Fig. 1 (a), users incorporate some text from personal database into the prompt as context (e.g., obtained by RAG (Lewis et al., 2020)). Ideally, the LLM should infer from this context that the user is currently located in Hong Kong and proceed to design a route from Hong Kong to Taipei. Concurrently, some small, segmented modules are deployed at the user's end (Zhou et al., 2023; Mai et al., 2024), to protect user privacy through the application of random perturbations to either embeddings or hidden states. On the server side, an adversary, while interactively providing LLM services, employs advanced inversion attack methods to reconstruct user's original data (Qu et al., 2021). The green box in Fig. 1 (a) indicates scenarios where the adversary is unable to reconstruct the data, signifying that privacy is preserved; conversely, the red box denotes situations where privacy is compromised. Fig. 1 (b) shows the goal of the defense *(i.e., the goal of this paper): server can still provide the accurate responses while being unable to obtain the privacy even using advanced attacks.*

### 3.2 EMPIRICAL STUDY OF PRIVACY VULNERABILITIES IN LLMS

In this part, we will illustrate through two interesting findings why it is challenging to effectively safeguard user data while maintaining the utility of LLMs, and without the in-depth analysis as well as the careful design, user privacy is quite vulnerable in cloud service scenarios.

#### 3.2.1 SPARSITY OF EMBEDDING SPACE

Currently, the tokenizer of open-source LLMs, represented by Llama (Dubey et al., 2024), has a vocabulary size of more than 100,000 tokens, while Gemma (Team et al., 2024) boasts a vocabulary

size of around 250,000 tokens. In the face of such a vast number of tokens, one might naturally inquire: *do the embeddings of these tokens cluster densely?* Contrary to this intuition, the embeddings of these tokens are, in fact, fairly sparsely distributed. In support of this, we design an experiment as follows. Considering the $(n-1)$-dimensional probability simplex whose vertices satisfy:

$$\left\{ w \in \mathbb{R}^n | \sum_{i=1}^{n} w_i = 1 \text{ and } w_i \geq 0 \text{ for } i = 1, \cdots, n \right\} \tag{1}$$

Obviously, if embedding space is very dense, when convex combinations with different weights $w_i$ are applied to different embeddings $E_i$ (where $E_i$ is the embedding vector of $i$-th token), the resulting new vectors $\sum_{i=1}^{n} w_i E_i$ are more likely to approximate other embeddings, rather than consistently maintaining the closest proximity to $\{E_i\}_{i=1}^{n}$. In light of this perspective, we randomly select embeddings from $n$ distinct tokens and subsequently sample weight $w$ from the $(n-1)$-simplex. For each vector $\sum_{i=1}^{n} w_i E_i$ resulting from the random convex combination of $\{E_i\}_{i=1}^{n}$, we identify the nearest token $\bar{T}$ (i.e., the embedding of $\bar{T}$ is closest to $\sum_{i=1}^{n} w_i E_i$) in the entire vocabulary list. By repeating this random process $N$ times, we calculate the average Inclusion Ratio (IR) as follows:

$$\text{IR} = \frac{1}{N} \sum_{k=1}^{N} \mathbb{I}_{\Theta^{(k)}}(\bar{T}^{(k)}), \tag{2}$$

where $\Theta^{(k)}$ is the set with $n$ tokens selected in the $k$-th round for the convex combination, and $\bar{T}^{(k)}$ is the identified nearest token in the $k$-th round. Indicator function $\mathbb{I}(\cdot)$ returns 1 if $\bar{T}^{(k)} \in \Theta^{(k)}$ else 0.

We set $N = 10,000$ for each $n$, and test on four open-source LLMs: Mistral (Jiang et al., 2023), Llama-3 (Dubey et al., 2024), Gemma-2 (Team et al., 2024) and Phi-3 (Abdin et al., 2024). Results are shown in Fig. 2. When $n \leq 8$, for all randomly sampled weights for convex combination, the token closest to the resulting vector is almost included within set $\Theta^{(k)}$. Furthermore, except for Gemma, such a phenomenon persists for the other three models when $n$ is increased to 32. We contend that these findings strongly demonstrate that the embedding space is sparse, as *a certain number of embeddings, combined convexly in any manner, do not approximate any other tokens except themselves.* This also implies a high degree of discriminability among the embeddings corresponding to distinct tokens.

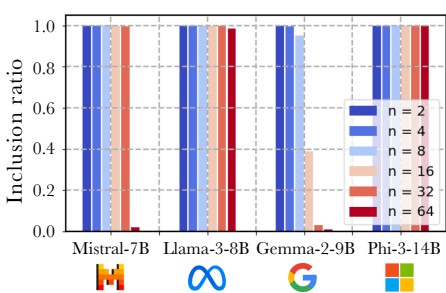

Figure 2: Inclusion ratio of resulting vector within the original token set, where each is statistically obtained on 10,000 experiments.

### 3.2.2 PRIVACY BREACHES FROM DIRECTIONS

Indeed, in the preceding part, we left an unaddressed issue: how to match a given vector (e.g., $\sum_{i=1}^{n} w_i E_i$ in above) to its nearest token. For an adversary, the fidelity of reconstructed tokens is directly impacted by this process. Consequently, we need to explore which methods are more prone to privacy breaches, as only then can we propose defensive strategies that are compelling. Unfortunately, this topic has not been comprehensively discussed in existing related work.

Typically, in distance measurement methods, two most commonly employed metrics are Euclidean distance and cosine distance. Prior studies (Qu et al., 2021; Zhang et al., 2024b) have predominantly considered the Euclidean distance for embeddings; however, in this section, we empirically demonstrate that the use of cosine distance is more advantageous for an adversary to match and reconstruct users' tokens with higher fidelity. To validate this, we randomly sample token embedding $E_i$ and introduce Laplacian noise with different scales of $\alpha \cdot \max(\text{abs}(E_i))$, where $\alpha \in \{0, 25, 0.5, 1, 2, 3, 4\}$. Subsequently, we employ Euclidean and cosine distance to match the perturbed embedding to its nearest token. After conducting 10,000 random trials, we calculate the proportion of tokens correctly recovered (i.e., the matched token is the original token), as detailed in Table 1.

In Table 1, regardless of the magnitude of noise scale, cosine matching consistently yields a higher proportion of correctly recovered tokens (hence, we employ it in the experiments of Fig. 1). Additionally, Table 1 corroborates the sparsity of the embedding space, demonstrating that even with the

Table 1: Proportion of correctly recovered tokens using Euclidean ($l_2$) and cosine ($cos$) distance matching metrics under Laplacian noise with scale of $\alpha \cdot \max\left(\text{abs}(E_i)\right)$.

|  | $\alpha = 0.25$ | | $\alpha = 0.5$ | | $\alpha = 1.0$ | | $\alpha = 2.0$ | | $\alpha = 3.0$ | | $\alpha = 4.0$ | |
|---|---|---|---|---|---|---|---|---|---|---|---|---|
|  | $l_2$ | $cos$ | $l_2$ | $cos$ | $l_2$ | $cos$ | $l_2$ | $cos$ | $l_2$ | $cos$ | $l_2$ | $cos$ |
| Mistral-7B-v0.3 | 1.00 | 1.00 | 1.00 | 1.00 | 0.99 | 1.00 | 0.57 | 0.93 | 0.09 | 0.45 | 0.02 | 0.14 |
| Llama-3-8B | 1.00 | 1.00 | 1.00 | 1.00 | 0.99 | 1.00 | 0.52 | 0.92 | 0.06 | 0.37 | 0.01 | 0.09 |
| Gemma-2-9B | 0.91 | 0.99 | 0.45 | 0.68 | 0.11 | 0.26 | 0.00 | 0.02 | 0.00 | 0.00 | 0.00 | 0.00 |
| Phi-3-14B | 1.00 | 1.00 | 1.00 | 1.00 | 1.00 | 1.00 | 0.58 | 0.99 | 0.17 | 0.66 | 0.03 | 0.26 |

introduction of random noise at a scale twice the size of the maximum absolute value (i.e., $\alpha = 2$), the original tokens can be recovered with a high success rate for Mistral, Llama and Phi (Gemma is lower due to its larger vocabulary size, leading embeddings more dense). Further, cosine distance is insensitive to the magnitude, a feature that is absent in Euclidean distance. Next, we will show the extreme vulnerability of privacy in LLMs under attacks based on cosine matching.

**Shallow layers of LLMs change direction slightly in embedding space.** Building upon the preceding findings, we now adopt the perspective of an adversary to propose a practical attack method. In this context, we do not consider the plaintext scenario (where users directly transmit data as prompts) but rather the scenario where users only send the hidden states $\mathbf{h} \in \mathbb{R}^{l \times d}$ to the server, where $l$ is the length of the tokenized prompt and $d$ is the size of hidden vector. The hidden states are derived from several attention layers deployed on the user's end, i.e., $\mathbf{h} = F(\mathcal{E}) = f_m \circ \cdots \circ f_2 \circ f_1(\mathcal{E})$, where $\mathcal{E}$ is the ordered set of token embeddings from user prompt and $f_i$ represents the $i$-th layer (Vaswani et al., 2017) in LLM. Then the optimization objective of the adversary can be expressed similarly to (Li et al., 2023):

$$\mathcal{E}^* = \arg\min_{\mathcal{E}'} \mathcal{L}\left(F(\mathcal{E}'), F(\mathcal{E})\right), \tag{3}$$

where $\mathcal{L}(\cdot)$ measures the distance between the reconstructed hidden states $\mathbf{h}'$ and the ground truth $\mathbf{h}$. Conventionally, we utilize gradient descent to update the dummy $\mathcal{E}'$ by minimizing the distance specified in (3), thereby obtaining the optimal $\mathcal{E}^*$. Subsequently, we apply the cosine matching, as previously introduced, to reconstruct tokens by the optimized $\mathcal{E}^*$. While we will later discuss the performance of this attack, we first pose an intriguing question: *What results might we obtain if we hypothesize $\mathcal{E}^* = \mathbf{h}$, followed by the direct application of cosine matching?* That is, we hypothesize that the user transmits hidden states $\mathbf{h}$, processed through $m$ attention blocks, to the server, while an adversary directly assumes $\mathcal{E}^* = \mathbf{h}$ and performs cosine matching to obtain $l$ tokens with the nearest directions to $\mathbf{h}$. We present experimental results for Llama in Table 2 (column "w/o"), reserving more in-depth analysis for the subsequent section, which will inform the development of our defense methods, and additional results for other models can be found in the Appendix C.1.

Table 2: Quantitative and qualitative results of attacks on Llama-3-8B with (column "opt") or without (column "w/o") gradient-based optimization as user employs $m$ attention layers.

|  | $m = 1$ | | $m = 5$ | | $m = 10$ | | $m = 15$ | | $m = 20$ | | $m = 25$ | |
|---|---|---|---|---|---|---|---|---|---|---|---|---|
|  | w/o | opt | w/o | opt | w/o | opt | w/o | opt | w/o | opt | w/o | opt |
| Rouge-1 | 1.00 | 1.00 | 0.96 | 1.00 | 0.88 | 0.91 | 0.67 | 0.93 | 0.40 | 0.84 | 0.23 | 0.84 |
| Rouge-2 | 1.00 | 1.00 | 0.93 | 1.00 | 0.73 | 0.84 | 0.50 | 0.82 | 0.25 | 0.69 | 0.04 | 0.69 |
| Rouge-L | 1.00 | 1.00 | 0.96 | 1.00 | 0.88 | 0.91 | 0.67 | 0.93 | 0.40 | 0.84 | 0.23 | 0.84 |

| *Truth* | Apple Inc is an American multinational corporation and technology company headquartered in Cupertino, California, in Silicon Valley. It is best known for its consumer electronics, software, and services. |
|---|---|
| m=10, w/o | Apple Inc is an American multinational corporation and technology company headquartered in Cupertinouk Californiashaw in Silicon Valley. It is best knownCA its consumer electronics, software,x and services. |
| m=10, opt | Apple Inc is an American multinational corporation and technology company headquartered in CupertinoGray Californiavirt in Silicon Valley. It is best known for its consumer electronics gating software0 and services. |
| m=25, w/o | Apple battalionstatesAn American Milton_testing bezTechnology companygrad_levelsDemon\Web plaza NOT vitamin Silicon,valueDean He reass best known tx consumerelectronics Gong software,$ produk,$Dean |
| m=25, opt | Apple Inc is an American multinational companies AND technology companies headquartered in Cupertino' California/ IN Silicon Valley. It is best known for its consumer electronics—for softwareTechnology and Services. |

We use Rouge (Lin, 2004) to assess the similarity between reconstructed and original texts. As shown in Table 2, even without any updates to $\mathcal{E}'$, the adversary can obtain nearly all private information

by user's hidden states **h** which is mapped through 10 attention blocks (blue text in Table 2). Such a result strongly suggests that the shallow layers of LLMs only minimally alter the direction in embedding space, thus making privacy susceptible to leakage. Moreover, when the adversary choose to optimize $\mathcal{E}'$ by gradient descent, even after passing through more layers, the essence of the original text is almost entirely reconstructed (see the last row in Table 2), which significantly underscores the vulnerability of privacy. The details about the attack implementation can be found in Appendix B.1.

### 3.3 PRIVACY ENHANCEMENT AND UTILITY COMPENSATION

In this section, we will first elucidate why the hidden states processed through multiple attention blocks can still directly leak privacy. Based on this understanding, we will design privacy-enhancing method to effectively resist adversarial reconstruction attacks.

Nowadays, mainstream decoder-based LLMs share a similar backbone. The architecture of transformer with residual blocks allows the model to break traditional constraints on the number of layers in neural networks, with the former providing scalability and the skip connections in the residual blocks enabling the training of very deep networks. The function of layer $i$ in decoder-based LLMs (refer to Fig. 3) can be mathematically expressed as follows (Vaswani et al., 2017). Note that we do not make a strict distinction between MHA and other attention mechanisms (e.g., GQA) here.

$$\mathbf{h}^- = \mathbf{h}^{(i-1)} + \underbrace{\text{MHA}\left(\text{RMSNorm}_1(\mathbf{h}^{(i-1)})\right)}_{\mathcal{J}_1}, \quad \mathbf{h}^{(i)} = \mathbf{h}^- + \underbrace{\text{FFN}\left(\text{RMSNorm}_2\left(\mathbf{h}^-\right)\right)}_{\mathcal{J}_2}, \quad (4)$$

where $\text{MHA}(\cdot)$ function as the multi-head attention block and $\text{FFN}(\cdot)$ function as the feed forward network. RMSNorm (Zhang & Sennrich, 2019) is adopted in nearly all mainstream LLMs due to its computational efficiency, which satisfies $\text{RMSNorm}(\mathbf{a}) = \mathbf{g} \odot \frac{\mathbf{a}}{\text{RMS}(\mathbf{a})}$, where $\mathbf{g}$ is the scaling parameters. We now make the conjectures to elucidate the circumstances under which the forward propagation of hidden states would significantly leak privacy.

**Proposition I.** *(Orthogonality) In the shallow layers, the cumulative sum of $\mathcal{J}_1 + \mathcal{J}_2$ is always located near the orthogonal subspace of token's embedding space.*

Appendix A.2 provides a validation and analysis for the proposition, which could reveal the underlying causes for the privacy vulnerabilities observed in different LLMs. Obviously, with Proposition I, even after forward propagation across several layers, the projections of hidden states in embedding space will barely be altered, leading to the direct leakage of privacy from the inner product-based cosine matching.

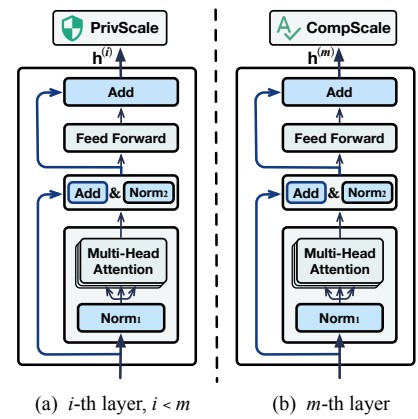

(a) *i*-th layer, *i* < *m*          (b) *m*-th layer

Figure 3: Architecture inside a transformer, where (a) PrivScale module is adpoted by user in the first $m-1$ layers and (b) CompScale module is adpoted in the $m$-th layer.

In the field of distributed learning, Ye et al. (2024) highlight from an optimization perspective that increasing the non-linearity of the model architecture will enhance the difficulty of privacy attacks. However, given the intricate nature of training LLMs, it is not feasible to redesign the model architecture and retrain from scratch. Consequently, the satisfactory defense must be plug-and-play. To achieve this requirement and effectively resist attacks, we propose to increase the proportion of $\mathcal{J}_1$ or $\mathcal{J}_2$ in Eq. (4), thereby amplifying the function of the nonlinear modules. However, adjusting $\mathcal{J}_1$ or $\mathcal{J}_2$ without careful consideration would undoubtedly severely impact the model's usability. Hence, we have designed a novel method which realizes the aforementioned objectives by shrinking each hidden state in $\mathbf{h}^{(i-1)}$ (i.e., $\mathbf{h}_j^{(i-1)} \in \mathbb{R}^d$, $j = 1, \cdots, l$) in a direction-preserving manner. This method offers two main benefits: first, after shrinking $\mathbf{h}_j^{(i-1)}$, the internal RMSNorm$_1$ of the MHA will restore it to its original scale, minimizing the impact on MHA's functionality; second, the shrinking of $\mathbf{h}^{(i-1)}$ will not alter the magnitudes of $\mathcal{J}_1$ and $\mathcal{J}_2$ significantly thanks to the normalization modules,

thus leading to $\mathbf{h}^-$ and $\mathbf{h}^{(i)}$ being more dominated by the non-linear structures. The theoretical analysis is provided in Appendix A.1, where it is demonstrated that our method causes the adversary's optimization objective less convex, making attacks harder to successfully implement.

Specifically, we apply a random scaling to the output of the first $i$-th layers (i.e., input of the $(i+1)$-th layer where $i < m$). Finally, we compensate for the shrinking by applying a direction-preserving amplification to the output of the $m$-th layer. Extensive experimental results will demonstrate that this form of direction-preserving scaling is effective in resisting attacks while guaranteeing usability of LLMs, including on several difficult tasks. The mathematical expression of our defense is given in the follows, where the output $\mathbf{h}^{(i)} \in \mathbb{R}^{l \times d}$ of $i$-th layer in Eq. 4 is re-expressed as:

$$\begin{cases} \mathbf{h}^{(i)} = (\mathbf{p}^{-1} \cdot \mathbf{1}_d^T) \odot \left[\mathbf{h}^- + \text{FFN}\left(\text{RMSNorm}_2\left(\mathbf{h}^-\right)\right)\right], & \text{if } i < m \\ \mathbf{h}^{(i)} = (\mathbf{c} \cdot \mathbf{1}_d^T) \odot \left[\mathbf{h}^- + \text{FFN}\left(\text{RMSNorm}_2\left(\mathbf{h}^-\right)\right)\right], & \text{if } i = m \end{cases} \quad (5)$$

where each entry in $\mathbf{p} \in \mathbb{R}^l$ is randomly sampled from the uniform distribution $p_j \sim U[1, 1+\delta]$ for each token in a context of length $l$. And $\mathbf{c} = c \cdot \mathbf{1}_l$ is a constant vector with compensation scalar $c$. In our experiments, scalar $c$ is obtained as follows: We select the first 20 of training data from the math task GSM8K (with CoT) and feed them into the privacy-enhanced inference model. Then we perform a simple search for scalar $c$ within a given range until we achieve the highest accuracy on these 20 math questions. This procedure is easy to execute and generally completes within a few minutes.

Overall, in our distributed inference paradigm designed to resist reconstruction attacks, a total of $m$ layers of privacy-enhancing and utility-compensating modules are deployed at the user-side. Further, in next section, we will show that low-bit quantization can be directly applied to these $m$ layers, without necessitating post-quantization calibrations.

## 4 EXPERIMENTS

### 4.1 IMPLEMENTATION SETTINGS

**Models, Tasks and Metrics.** We use five instructed models to evaluate our method, including Mistral-7B-v0.3, Llama-3-8B, Gemma-2-9B, Phi-3-14B and Llama-3-70B-AWQ, and use six tasks for different privacy-preserving evaluations. Specifically, we protect all context for HellaSwag (Zellers et al., 2019), BoolQ (Clark et al., 2019), GSM8K (Cobbe et al., 2021) and HumanEval (Chen et al., 2021). In addition, we protect few-shot examples like Tang et al. (2024) for tasks which employ few-shot learning, including MMLU (Hendrycks et al., 2021) and BBH (Suzgun et al., 2022). In Appendix B.3, we present a clear depiction of the protected part in these tasks and encourage readers to review. For evaluating the attack (with optimization), we use Rouge-1, Rouge-2 and Rouge-L (Lin, 2004), where Rouge-1 measures the word-level (1-gram) reconstruction capability while Rouge-2 measures phrase-level (2-gram) and Rouge-L measures Longest Common Subsequence (LCS).

**Criteria for Parameter Selection.** We investigate the influence of $\delta$ for $\mathbf{p}$ in (5) on the quality of the reconstructions (we can search for the appropriate $\delta$ through conducting attack and defense locally by the $m$ local layers). We use contexts in typical reading comprehension task (BoolQ) as targets and the statistical results are shown in Fig. 4 (a). Fig. 4 (b) proves that with the conditions of Rouge-1 $< 0.5$, Rouge-2 $< 0.3$, Rouge-L $< 0.5$, it is sufficient for the reconstruction to compromise a significant amount of privacy information from the original data (more results are in Appendix C.3). According to this, as well as the results in Fig. 4 (a), we set $\delta$ to $[0.30, 0.20, 0.35, 0.50, 0.425]$ for Mistral-7B-v0.3, Llama-3-8B, Gemma-2-9B, Phi-3-14B and Llama-3-70B-AWQ, respectively.

Taking into account the requirement to counteract an adversary's random guessing, as well as the computational capabilities of user devices, we have configured the number of local layers $m = 10$. With a total of 9 (i.e., $m - 1$) consecutive layers, each accompanied by a distinct random scaling transformation applied to the hidden states corresponding to every token (and re-randomized for each inference), we believe this setup is sufficient to prevent an adversary from accurately guessing the specific scaling magnitude applied to the victim's data. As for the compensation scalar $c$, the results of rough search are shown in Fig. 4 (c), and based on this, the employed $c$ is $[1.5, 1.0, 1.5, 2.0, 2.0]$, respectively. More about the experimental setup of Fig. 4 is given in the Appendix B.2. And in the future, we will delve into the investigation of more refined strategies for noise insertion based on the degree of module non-linearity, as well as explore configurations with smaller $m$.

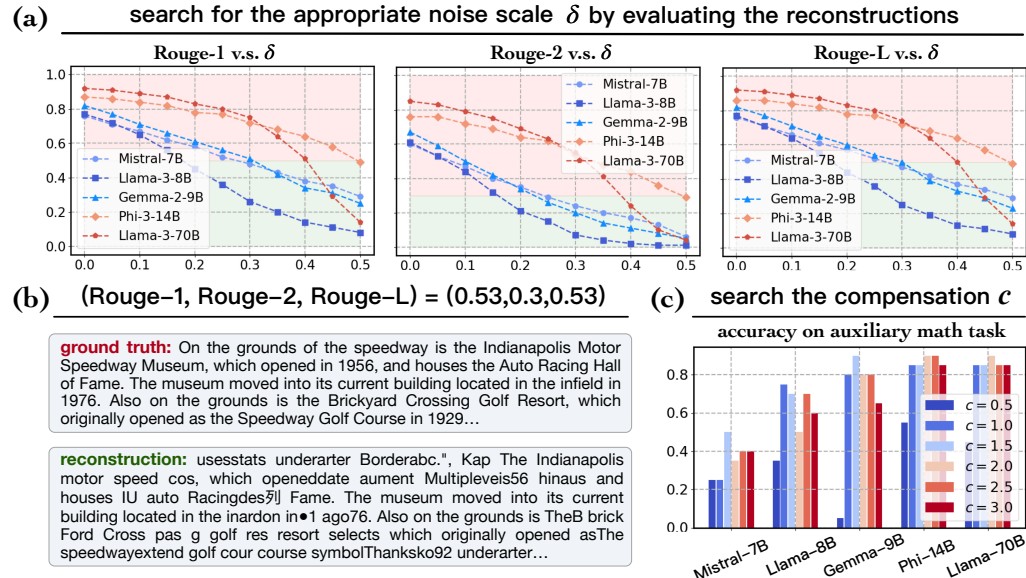

Figure 4: Algorithm parameters selection, where (a) illustrates the Rouge scores with different noise scale $\delta$, with Rouge-1 < 0.5, Rouge-2 < 0.3, Rouge-L < 0.5 considered as privacy thresholds in this paper; (b) presents an attack result with (Rouge-1, Rouge-2, Rouge-L)=(0.53,0.3,0.53); (c) shows the accuracies on math task (first 20 training data of GSM8K) with different compensation $c$.

## 4.2 RESISTING ATTACKS

In this part, we assess the proposed method on resisting reconstruction attacks. The quantitative results are presented in Table 3, and the qualitative results are given in Fig. 5. More experimental results are given in Appendix C.2 and C.3, including the attack results without countermeasure, as well as resisting attacks across various contexts from different datasets.

In Table 3, all Rouge scores meet the criteria outlined in the previous part. Furthermore, as indicated in Fig. 5, our proposed defense method significantly safeguards a substantial amount of private information for all LLMs, even in cases (e.g., Llama-3-70B and Phi-3-14B) where, the Rouge-1 and Rouge-L scores of these reconstructions slightly over 0.5. These results substantiate the efficacy of our method in resisting attacks.

Table 3: Rouge scores when using defense.

|  | Rouge-1 | Rouge-2 | Rouge-L |
|---|---|---|---|
| Mistral-7B | 0.48 | 0.24 | 0.47 |
| Llama-3-8B | 0.45 | 0.21 | 0.44 |
| Gemma-2-9B | 0.42 | 0.14 | 0.39 |
| Phi-3-14B | 0.49 | 0.29 | 0.49 |
| Llama-3-70B | 0.39 | 0.17 | 0.39 |

*Ground Truth:* On the grounds of the speedway is the Indianapolis Motor Speedway Museum, which opened in 1956, and houses the Auto Racing Hall of Fame. The museum moved into its current building located in the infield in 1976. Also on the grounds is the Brickyard Crossing Golf Resort, which originally opened as the Speedway Golf Course in 1929…

*Llama-3-8B:* on the fastest of the speed bridge townrepositories Indianapolis Motor Speedway Museum-city which open open open197 Fen("../../ andINST刊 Pratt mathematics Hall堂 Museumexists The museum moving into its current building located in the infield_begin こ こ197ptaZW also on⊑ campus☺۷ the brick Land瀬 golfформа zg which originally open as the Speedway proliferationHospitalVi notes192_PORT>')… *R1,R2,RL = 0.48,0.26,0.48*

*Llama-3-70B-AWQ:* On On grounds grounds the speed Speedway Conditions Homemade Indianapolis Motor Speedway Museum museum which případech případech případech19556.getOwnPropertyDescriptor and houses prostřednictvím Auto Racing Hallphi Fame fascination_$_ museum moved into эта current building located intέρα infield غرона prostřednictvím̌ä Also on电 grounds is Kron Brickyard Crossing Golf Resort Neo which originallyRay případech 음악 Speedway_det-rays_det☼शक19229운… *R1,R2,RL = 0.52,0.26,0.52*

*Mistral-7B-v0.3:* on groundsrian Park speedway is toile Indian Chicago motor MountCanvas Museum Court which opened in two COVID risks6, and houses the auto Racing Hall Jah Museumrom The Museum moved into its current buildingsodio in ArabiaDishof inansionapis266. also ионлъ染 is TheB bacteriayal Crossler Golfsrops, which × started as the Square Square Golf incidentOpenrom stagiverseraid Normem… *R1,R2,RL = 0.46,0.20,0.46*

*Gemma-2-9B:* On grounds grounds local speed Speedwayesta MD Indianapolis Indianapolis Speedway Museumcock museum opensgb Stadiumley park RBTC park associate housed Ver Auto Racing Hall Hall Fame blog CN museum Move ONcam Current Building located park Area infield azgeo Info win NL reminiscent Centre Also ON circuit Grounds Cer Centre Brickyard Crossing Golf golf park plus originally opens as National Speedway Golf Golf in Ela citypro golfmers… *R1,R2,RL = 0.34,0.15,0.34*

*Phi-3-14B:* usesstats underarter Borderabc.", Kap The Indianapolis motor speed cos, which openeddate aument Multipleveis56 hinaus and houses IU auto Racingdes列 Fame. The museum moved into its current building located in the inardon in●1 ago76. Also on the grounds is TheB brick Ford Cross pas g golf res resort selects which originally opened asThe speedwayextend golf cour course symbolThanksko92 underarter… *R1,R2,RL = 0.53,0.30,0.53*

Figure 5: Reconstructions of the attack on LLMs equipped with our defense. Best viewed zoomed in.

## 4.3 IMPACT ON UTILITY

We now need to consider whether a LLM can still function effectively after the "protection" of critical information, particularly in tasks involving math or code where content such as numbers and variables are decisive for the answers. Consequently, we have to deeply evaluate the remaining performance of models equipped with the proposed defense mechanism across various tasks. Simultaneously, we investigate the impact on model performance of directly perturbing embeddings or replacing tokens by nearest neighbors (see Appendix C.4 for details), with experimental results indicating that these strategies severely compromise performance, especially in coding and mathematical tasks, even when the perturbation scale is insufficient to counter reconstruction attacks.

**Choice-based tasks.** Choice-based tasks involve choosing the correct answer from multiple choices (here we consider BoolQ (Clark et al., 2019) as a choice-based task, despite its responding with True or False rather than an explicit choice). In HellaSwag (commonsense reasoning, 0-shot) and BoolQ (reading comprehension, 0-shot), we apply privacy-preserving defenses to all context, which serves as the direct basis for the model's responses. In MMLU (57 subjects, 1-shot for Llama-70B and 5-shot for others), we treat all examples as privacy like Tang et al. (2024) and protect them. Experimental results are presented in Table 4. For all experiments within the same task, we use the same prompts. Obviously, after applying defense, LLMs maintain quite good performance across these choice-based tasks. We also showcase the performance of LLMs across four subcategories of the MMLU. The results indicate that our method will not significantly degrade the performance of LLMs on a particular category.

Table 4: Accuracies of different tasks, where: "w/o" not using defense, "def" using defense.

|  | HellaSwag | | BoolQ | | MMLU | | ◇ STEM | | ◇ Human | | ◇ Social | | ◇ Other | |
|---|---|---|---|---|---|---|---|---|---|---|---|---|---|---|
|  | w/o | def | w/o | def | w/o | def | w/o | def | w/o | def | w/o | def | w/o | def |
| Mistral-7B | 66.3 | 61.7 | 85.1 | 82.9 | 60.1 | 59.4 | 48.8 | 49.3 | 57.4 | 56.0 | 69.3 | 68.3 | 66.7 | 66.0 |
| Llama-8B | 66.7 | 65.4 | 84.3 | 83.0 | 65.8 | 65.2 | 55.8 | 54.6 | 60.9 | 60.5 | 76.0 | 75.5 | 73.3 | 72.9 |
| Gemma-9B | 81.9 | 80.1 | 89.2 | 87.7 | 72.2 | 72.1 | 65.7 | 65.0 | 66.1 | 67.2 | 83.5 | 82.7 | 76.8 | 76.5 |
| Phi-14B | 89.8 | 87.0 | 88.7 | 85.2 | 76.9 | 75.3 | 69.5 | 68.2 | 73.4 | 70.7 | 85.8 | 84.9 | 80.9 | 80.0 |
| Llama-70B | 85.1 | 83.0 | 89.7 | 83.2 | 77.7 | 74.0 | 71.6 | 70.5 | 72.8 | 67.3 | 86.6 | 82.2 | 82.4 | 79.8 |

**Non choice-based tasks.** In this part, we evaluate model's performance on the math task GSM8K (0-shot, with CoT) and the code task HumanEval (0-shot, pass@1). We apply protection directly to the context upon which all responses in GSM8K and HumanEval rely (see Appendix B.3). Results are presented in Table 5. Compared to choice-based tasks, there is a slightly greater performance decline in math and coding tasks, due to these tasks being more granular in nature and we have protected all their contexts. Even so, these LLMs remain effective, as that even after applying defense, their performance is either superior or comparable to that of slightly smaller models.

Table 5: Accuracies under different settings: "d-8" and "d-4" for defense with 8-bit and 4-bit quantization, "$L(\alpha)$" for perturbing embeddings following $\alpha = 0.5$ in Table 1, "NR" for nearest replacing, which performs extremely worse than BoolQ on math task GSM8K and code task HumanEval.

|  | GSM8K (0-shot, CoT) | | | | | | HumanEval (pass@1) | | | | | | BoolQ (0-shot) | | | |
|---|---|---|---|---|---|---|---|---|---|---|---|---|---|---|---|---|
|  | w/o | def | d-8 | d-4 | $L(\alpha)$ | NR | w/o | def | d-8 | d-4 | $L(\alpha)$ | NR | d-8 | d-4 | $L(\alpha)$ | NR |
| Mistral-7B | 54.8 | 46.8 | 46.6 | 43.4 | 2.1 | 3.5 | 38.4 | 34.1 | 39.0 | 38.4 | 5.5 | 3.0 | 82.6 | 82.8 | 42.6 | 73.1 |
| Llama-8B | 77.8 | 72.6 | 73.2 | 70.7 | 2.0 | 5.5 | 55.5 | 51.2 | 50.0 | 47.6 | 0 | 0 | 82.7 | 81.1 | 56.6 | 76.5 |
| Gemma-9B | 86.4 | 84.3 | 84.5 | 85.8 | 1.7 | 3.8 | 63.4 | 58.5 | 57.3 | 57.3 | 0 | 21.3 | 87.9 | 87.8 | 64.2 | 72.8 |
| Phi-14B | 91.1 | 85.2 | 85.2 | 78.1 | 2.3 | 4.5 | 70.1 | 64.6 | 63.4 | 58.5 | 4.9 | 4.3 | 84.7 | 82.8 | 70.5 | 76.5 |
| Llama-70B | 92.9 | - | - | 86.4 | 1.5 | 7.2 | 78.7 | - | - | 71.3 | 1.2 | 2.4 | - | 83.2 | 45.0 | 84.9 |

**Impact on few-shot learning.** BBH evaluates models using few-shot examples, and these examples are crucial as they determine how LLMs organize chain-of-thought and generate responses. Different from previous experiments, in this part, we demonstrate that for tasks where the performance is better with 3-shot compared to 1-shot (not all tasks benefit from more examples), the addition of defense to all 3 examples still yields superior performance over 1-shot without defense. This experiment is designed to show that even with defense, LLMs can still effectively learn knowledge from examples.

To this end, we only evaluate on a subset of tasks from the BBH where 3-shot outperforms 1-shot (details are in Appendix C.5). Obviously, in Table 6, after applying defense, these LLMs still "learn" examples effectively and outperform those using 1-shot learning without defense. Owing to the lengthy computation time, we only evaluate the first 20 questions for each task in BBH for Llama-70B, and this setting does not affect the analysis.

Table 6: Accuracies on selected tasks in BBH.

| | **BIG-Bench Hard** (CoT) | | |
| | w/o(3-shot) | def(3-shot) | w/o(1-shot) |
| --- | --- | --- | --- |
| Mistral-7B | 55.0 | 52.7 | 46.7 |
| Llama-8B | 68.2 | 67.5 | 57.4 |
| Gemma-9B | 77.8 | 75.6 | 71.2 |
| Phi-14B | 73.5 | 68.3 | 61.6 |
| Llama-70B | 77.7 | 72.3 | 63.2 |

**Impact of quantization, perturbation and replacement.** In this part, we select three representative tasks—math, coding and reading comprehension—to investigate the influence of applying low-bit quantization to the user-side modules when using our defense (see Table 5, note that the Llama-70B we used is downloaded from Hugging Face (Wolf et al., 2020) and is already quantized to 4-bit by AWQ). We also evaluate the impact on model utility by introducing perturbations to the embeddings, as well as replacing each token with its nearest token in embedding space (column "NR" in Table 5).

In Table 5, using our defense with 8-bit quantization will not significantly compromise model performance further. However, when using 4-bit quantization, there may be a noticeable performance degradation on a few tasks (in red). In contrast, for the way of perturbing embeddings, we use the setting with $\alpha = 0.5$ as in Table 1, which almost completely fails to protect privacy, yet significantly degrades usability, particularly in math and coding tasks. As for the nearest replacing, a similar result is observed, which is comprehensible, as the performance of math and coding tasks is contingent upon token-level granularity, whereas replacing tokens with the nearest neighbors has a relatively smaller influence on text comprehension (comparison before and after nearest replacing is in Appendix C.4).

We also report the runtime GPU memory required by the user when using different quantization precisions (see Table 7). We apply HQQ quantization (Badri & Shaji, 2023) to all 10 local layers except for Llama-70B-AWQ, which is already quantized by AWQ (Lin et al., 2024). These 10 layers' required GPU memory is shown in the middle part of Table 7. The embedding layer of LLMs primarily involves memory access operations rather than dense floating-point computations, therefore, whether to transfer it to GPU memory is optional.

Table 7: Memory required by the user in GB, "embed" for embedding layer's memory.

| | FP/BF16 | 8-bit | 4-bit | embed |
| --- | --- | --- | --- | --- |
| Mistral-7B | 4.06 | 2.03 | 1.02 | 0.25 |
| Llama-8B | 4.06 | 2.03 | 1.02 | 0.98 |
| Gemma-9B | 3.69 | 1.85 | 0.92 | 1.71 |
| Phi-14B | 6.35 | 3.17 | 1.59 | 0.31 |
| Llama-70B | - | - | 4.14 | 1.96 |

In Table 7, even the 70B model requires a memory size which is affordable for mobile devices. With the advancement of on-device AI and the development of flagship AI chips (Tan & Cao, 2021; Gerganov et al., 2023), we believe that the proof-of-concept proposed in this paper will help to achieve a balance between privacy, utility, and memory efficiency for the future of on-device AI.

## 5 CONCLUSION AND FUTURE WORK

This paper exposes the significant vulnerability of user privacy when employing LLM cloud services, and we contend that the attack method employed herein can serve as a benchmark for related research. Meanwhile, to alleviate the privacy leakage, we introduce a plug-and-play distributed inference paradigm. Extensive experimental results have demonstrated that our method can effectively resist privacy attacks while maintaining the usability of the model.

However, our work has several limitations. Firstly, the coarse-grained nature of our privacy-preserving shrinking operation on hidden states could be improved. Actually, a more granular strategy could be designed based on the sequence length (hidden states closer to the end of the sequence are more impacted due to the cumulation of preceding hidden states) and the non-linearity of modules, which would further mitigate the compromise on model performance. Additionally, in a few scenarios, performance degration may occur after directly quantizing model to 4-bit, where post-quantization calibration might be helpful (Frantar et al., 2022). Moreover, our method requires local-server collaboration for inference, implying the local device must have some computational capability. We will focus on addressing these limitations in our future work.

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

# A ADDITIONAL ANALYSIS

## A.1 BASIS FOR THE DEFENSE

Due to the non-convexity and parameter complexity of deep neural networks, the analysis of even simple two-layer nonlinear networks for traditional machine learning problems such as learning halfspaces heavily relies on prior assumptions. Consequently, we here simplify the analysis of attack and defense without compromising the final conclusions, i.e., our approach will render attacks more difficult to succeed.

Firstly, we simplify a part of layer functionality to $\mathbf{h}(\mathbf{x}) = \zeta \cdot \mathbf{x} + \mathbf{f}(\mathbf{x})$, where $\mathbf{x}$ represents the input data, $\mathbf{f}$ is the nonlinear module within this layer, $\zeta$ is a constant term, and the addition comes from the skip connections in the residual block. Note that we can use $\mathbf{x}$ instead of $\zeta \cdot \mathbf{x}$ as the input for function $\mathbf{f}(\cdot)$, thanks to the capability of $\text{RMSNorm}_1$ (see Eq. 4), which enables that the input for $\mathbf{f}(\cdot)$ is not affected by the scaling operation. We now simply assume the attacker's objective function to be $l(\mathbf{x}) = \frac{1}{2}\|\mathbf{h}(\mathbf{x}) - \mathbf{h}(\mathbf{t})\|_2^2$, where $\mathbf{t}$ is the target data. The attacker needs to iteratively optimize $\mathbf{x}$ to minimize the $l(\mathbf{x})$. Our proof objective is to demonstrate that *as $\zeta$ increases, the optimization objective function $l(\mathbf{x})$ becomes closer to a convex function, thus possessing a more favorable optimization landscape, which facilitates the convergence of $\mathbf{x}$ to $\mathbf{t}$.*

*Sketch of Proof.* For $\mathbf{h}(\mathbf{x}) = \zeta\mathbf{x} + \mathbf{f}(\mathbf{x})$ and $l(\mathbf{x}) = \frac{1}{2}\|\mathbf{h}(\mathbf{x}) - \mathbf{h}(\mathbf{t})\|_2^2$, we have:

$$\nabla l(\mathbf{x}) = \mathbf{J}_\mathbf{h}^T \left[\mathbf{h}(\mathbf{x}) - \mathbf{h}(\mathbf{t})\right] = \left[\zeta \cdot \mathbb{I}_{d \times d} + \mathbf{J}_\mathbf{f}^T\right] \left[\zeta\mathbf{x} + \mathbf{f}(\mathbf{x}) - \zeta\mathbf{t} - \mathbf{f}(\mathbf{t})\right], \quad (6)$$

here we simplify the dimension of $\mathbf{x}$ to $d$, and $\mathbb{I}_{d \times d}$ is an identity matrix, $\mathbf{J}_\mathbf{h}$ and $\mathbf{J}_\mathbf{f}$ are Jacobian matrixes corresponding to $\mathbf{h}(\mathbf{x})$ and $\mathbf{f}(\mathbf{x})$. Then the Hessian of the attack objective $l(\mathbf{x})$ can be calculated as:

$$\mathbf{H}_l = \left(\zeta \cdot \mathbb{I}_{d \times d} + \mathbf{J}_\mathbf{f}^T\right)\left(\zeta \cdot \mathbb{I}_{d \times d} + \mathbf{J}_\mathbf{f}\right) + \sum_{i=1}^{d} \left[\zeta\mathbf{x} + \mathbf{f}(\mathbf{x}) - \zeta\mathbf{t} - \mathbf{f}(\mathbf{t})\right]_i \cdot \mathbf{H}_{\mathbf{f}_i}$$

$$= \left[\zeta^2 \cdot \mathbb{I}_{d \times d} + \mathbf{J}_\mathbf{f}^T\mathbf{J}_\mathbf{f} + \zeta(\mathbf{J}_\mathbf{f} + \mathbf{J}_\mathbf{f}^T)\right] + \sum_{i=1}^{d} \underbrace{\left[\zeta\mathbf{x} + \mathbf{f}(\mathbf{x}) - \zeta\mathbf{t} - \mathbf{f}(\mathbf{t})\right]_i \cdot \mathbf{H}_{\mathbf{f}_i}}_{\mathbf{T}_i}, \quad (7)$$

Notice that $\mathbf{H}_l$ contains a term $\zeta^2$, which means that as $\zeta$ increases, this term will significantly contribute to the $\mathbf{H}_l$. Since $\zeta^2 > 0$, this will make the $\mathbf{H}_l$ more likely to be positive definite (which is the key property of $l(\mathbf{x})$ being convex), as its each eigenvalue satisfies:

$$\lambda_k(\mathbf{H}_l) = \lambda_k \left( \left[\mathbf{J}_\mathbf{f}^T\mathbf{J}_\mathbf{f} + \zeta(\mathbf{J}_\mathbf{f} + \mathbf{J}_\mathbf{f}^T)\right] + \sum_{i=1}^{d} \underbrace{\left[\zeta\mathbf{x} + \mathbf{f}(\mathbf{x}) - \zeta\mathbf{t} - \mathbf{f}(\mathbf{t})\right]_i \cdot \mathbf{H}_{\mathbf{f}_i}}_{\mathbf{T}_i} \right) + \zeta^2, \quad (8)$$

where $\lambda_k(\cdot)$ represents $k$-th eigenvalue of $(\cdot)$, and $\zeta^2$ directly contributes to it. Additionally, $\mathbf{H}_{\mathbf{f}_i}$ is the Hessian of $[\mathbf{f}(\mathbf{x})]_i$, and since $\mathbf{f}(\mathbf{x})$ in neural network is usually non-convex, $\mathbf{H}_{\mathbf{f}_i}$ as well as the tensor $\mathbf{T}_i$ in Eq. (8) are usually not positive semi-definite. However, as $\zeta$ increases, the $\zeta^2$ term will dominate in the Hessian $\mathbf{H}_l$, thus "masking" the non-convex nature from $\sum_{i=1}^{d} \mathbf{T}_i$ and $(\mathbf{J}_\mathbf{f} + \mathbf{J}_\mathbf{f}^T)$.

Now we return to our defense. Based on the above conclusion, when we inversely scale down $\zeta$ (i.e, $\mathbf{p}^{-1}$ in Eq. 4), the Hessian $\mathbf{H}_l$ is more likely to be dominated by non-positive definite terms. This, in turn, makes attacker's objective more prone to deviate from convexity, deteriorating the optimization landscape, ultimately making it harder for the attack to converge to the target data $\mathbf{t}$. $\qquad\square$

In fact, the above assumption $\mathbf{h}(\mathbf{x})$ refers to the network before the FFN layer. However, it is not difficult to infer that if we are impossible to reconstruct the original data $\mathbf{t}$ from $\mathbf{h}(\mathbf{t})$, then we are also impossible to reconstruct $\mathbf{t}$ from $\text{NN}(\mathbf{h}(\mathbf{t}))$ (NN represents the deeper parts of the network), since the reconstruction process is propagated layer by layer in reverse. That is, to correctly reconstruct $\mathbf{t}$ from $\text{NN}(\mathbf{h}(\mathbf{t}))$, one must implicitly and correctly infer $\mathbf{h}(\mathbf{t})$ first, and then could they correctly infer $\mathbf{t}$ by $\mathbf{h}(\mathbf{t})$ implicitly. If it is hard to infer $\mathbf{t}$ from $\mathbf{h}(\mathbf{t})$, it is evident that the attacker would also be unable to reconstruct $\mathbf{t}$ from $\text{NN}(\mathbf{h}(\mathbf{t}))$. Our defense strategy essentially involves applying the aforementioned attack-hardening measures to $m - 1$ sub-modules within the network, thereby providing a certain level of privacy safeguarding in optimization perspective.

## A.2 Basis for the Proposition

This part demonstrates that the cumulative sum of $\mathcal{J}_1 + \mathcal{J}_2$ (see Eq. 4) across shallow layers is always located near the orthogonal subspace of token's embedding space. Clearly, according to Eq. 4, the output for each layer satisfies the following form: $\mathbf{h}^{(i)} = \mathbf{E} + \text{NN}^{(i)}(\mathbf{E})$, where $\mathbf{E}$ is the embeddings of input tokens, $\text{NN}^{(i)}(\cdot)$ mimics the functionality of the first $i$ layers of the network, and this form holds thanks to the residual structure within the network. Therefore, we calculated the angle between $\text{NN}^{(i)}(\mathbf{E})$ (i.e., the cumulative sum of $\mathcal{J}_1 + \mathcal{J}_2$) and the embedding space for the shallow layers, in order to verify that the LLMs primarily function in the orthogonal subspace of the embedding space.

Specifically, to estimate the angle between $\text{NN}^{(i)}$ and the embedding space, we randomly sample 1,000 tokens and input them to LLMs to obtain the set $\Psi_i = \left\{ \text{NN}^{(i)}(\mathbf{E}_j) \right\}_{j=1}^{1,000}$ from layer $i$. At the same time, we randomly chose 10,000 tokens and use their embeddings to construct set $\Phi = \{E_k\}_{k=1}^{10,000}$. Finally, we calculate the average angles of each element in $\Psi_i$ with respect to all elements in $\Phi$. Results are shown in Fig. 6.

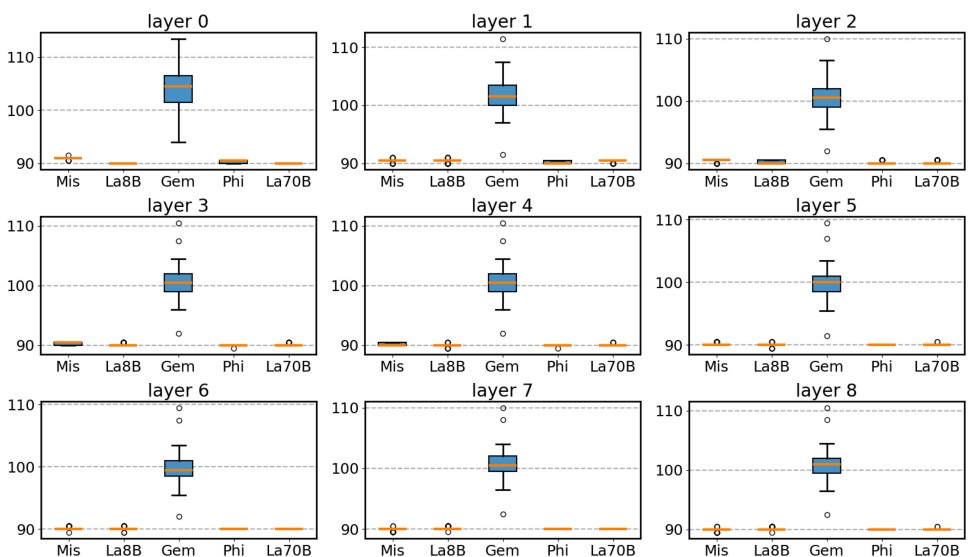

Figure 6: Distribution of angles between $\text{NN}^{(i)}(\mathbf{E})$ and the embeddings, y-axis unit: degrees ($^\circ$).

Note that the results of Gemma differ slightly from the other models. We speculate that this is because in the decoders of Gemma, additional $\text{RMSNorm}(\cdot)$ are applied to $\mathcal{J}_1$ and $\mathcal{J}_2$ in each layer. However, it is clear that the angles between $\text{NN}^{(i)}(\mathbf{E})$ and token embeddings are centered near 90 degrees, which leads to small projection for $\text{NN}^{(i)}(\mathbf{E})$ in the embedding space (even for the 100-degree projection of $\text{NN}^{(i)}(\mathbf{E})$ in Gemma). In other words, the projection of $\mathbf{h}^{(i)}$ in the embedding space changes very little. Furthermore, based on our previous findings, the sparsity of the embedding space leads the attack robust to certain perturbations (also sufficient to cope with the 100-degree projection of $\text{NN}^{(i)}(\mathbf{E})$ from Gemma), allowing an attacker to easily match the original tokens in the embedding space based on $\mathbf{h}^{(i)}$.

# B More Experimental Configurations

## B.1 Attack Implementation

For the optimization-based attack, we use Adam optimizer to iteratively update $\mathcal{E}'$ in Eq. (3) with an initial learning rate of 0.01. We perform a total of 200 optimization steps for each attack, and apply linear decay to the learning rate, with a minimum learning rate of 0.002. For the Adam optimizer, we set $\beta_1 = 0.9, \beta_2 = 0.999$, and use the default settings for all other parameters. As for the distance

function $\mathcal{L}(\cdot)$ in (3), since our defense method employs a direction-preserving random scaling transformation, meaning the amplitude of the target vectors is randomly altered, using Euclidean distance as the objective function for the attack obviously has significant bias. Therefore, we use the hidden state-level cosine distance $\mathcal{L}\left(F(\mathcal{E}'), F(\mathcal{E})\right) = \frac{1}{l} \sum_{i=1}^{l} \left[1 - \frac{\langle F_i(\mathcal{E}'), F_i(\mathcal{E}) \rangle}{|F_i(\mathcal{E}')||F_i(\mathcal{E})|}\right]$ as the objective function ($l$ is the length of the sequence, and $F_i(\mathcal{E}'), F_i(\mathcal{E})$ are hidden states corresponding to $i$-th token), which inherently has amplitude robustness, thereby achieving a higher-performance attack. Note that using this objective function does not impede deriving conclusions similar to those of A.1.

## B.2 SETTINGS FOR PARAMETER SELECTION

For the experiment in Fig. 4(a), we use the method introduced in B.1 as attack and BoolQ as the target data, and gradually increase the noise scale $\delta$ within the range of 0 to 0.5 and recording the corresponding average attack performance. We select the case that is closest to the privacy threshold to display in Fig. 4(b). After selecting appropriate noise scales for all models based on Fig. 4(a), we search for the optimal compensation coefficient $c$ for layer $m$ in the range of 0.5 to 3.0, choosing the one where the model achieved the highest accuracy on the first 20 training samples in GSM8K after being compensated with this coefficient. The results are presented in Fig. 4(c).

## B.3 PROTECTED PART FOR DIFFERENT TASKS

In Fig. 7, we present a part of prompts for different tasks, along with the parts where we apply privacy protection (in the green boxes).

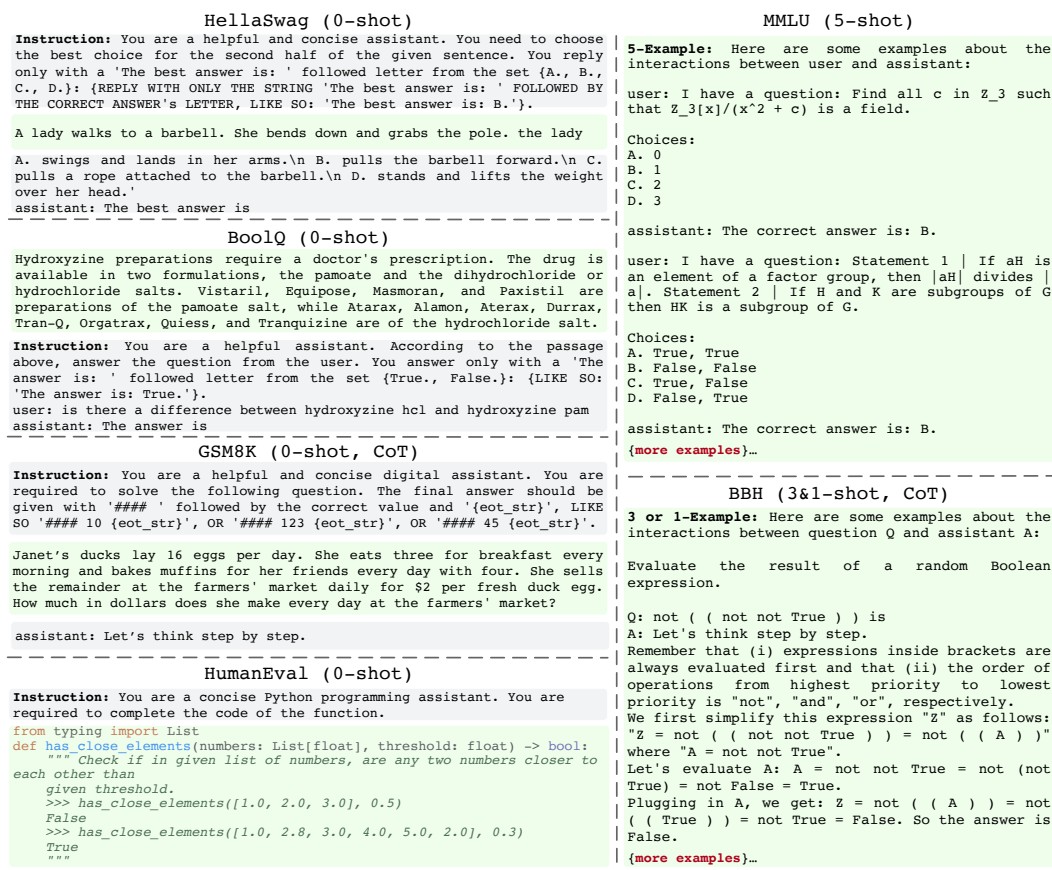

Figure 7: Prompt templates tailored for different tasks. The green boxes represent the parts where we apply privacy protection. For the tasks on the left, we protect all critical contexts that can directly determine the answer of LLMs. For the right part, we protect all examples like Tang et al. (2024).

## C    MORE RESULTS

### C.1    ATTACK RESULTS WITH AND WITHOUT OPTIMIZATION

We give more results of the attack on Mistral (Table 8), Gemma (Table 9), Phi (Table 10) and Llama (Table 11) with or without optimization. Note that we did not use any defensive measures in this part.

Table 8: Quantitative and qualitative results of attacks on Mistral-v0.3 with or without optimization.

|  | $m = 1$ | | $m = 5$ | | $m = 10$ | | $m = 15$ | | $m = 20$ | | $m = 25$ | |
|---|---|---|---|---|---|---|---|---|---|---|---|---|
|  | w/o | opt | w/o | opt | w/o | opt | w/o | opt | w/o | opt | w/o | opt |
| Rouge-1 | 1.00 | 1.00 | 0.81 | 0.88 | 0.75 | 0.81 | 0.75 | 0.75 | 0.70 | 0.73 | 0.66 | 0.67 |
| Rouge-2 | 1.00 | 1.00 | 0.70 | 0.81 | 0.61 | 0.70 | 0.59 | 0.62 | 0.55 | 0.55 | 0.50 | 0.50 |
| Rouge-L | 1.00 | 1.00 | 0.81 | 0.88 | 0.75 | 0.81 | 0.75 | 0.75 | 0.70 | 0.73 | 0.66 | 0.67 |

| | |
|---|---|
| *Truth* | Apple Inc is an American multinational corporation and technology company headquartered in Cupertino, California, in Silicon Valley. It is best known for its consumer electronics, software, and services. |
| m=10, w/o | Apple Inc is an American mult world International corporation and technology company head headquarters'orte in Cu Appleville, California, in Silicon Valley. It is best known for its consumeronics, software, and services |
| m=10, opt | Apple Inc is an American mult internation International corporation and technology company head headquarters' sede in Cu Appleino, California, in Silicon Valley. It is best known for its consumer electronics, software, and services |
| m=25, w/o | Apple Inc is an American multin entity Corporation and technology company head02ized inuptdale. California, in Sil SilUn it is best known for its consumeron e, Software, and servicesnik |
| m=25, opt | Apple Inc is AN American mult internation corporation corporation and technology company headmq aged in Ca Russonal, California, in Sil Sil Valley. it is best known for its consumer electronattle, software, and services. |

Table 9: Quantitative and qualitative results of attacks on Gemma-2-9B with or without optimization.

|  | $m = 1$ | | $m = 5$ | | $m = 10$ | | $m = 15$ | | $m = 20$ | | $m = 25$ | |
|---|---|---|---|---|---|---|---|---|---|---|---|---|
|  | w/o | opt | w/o | opt | w/o | opt | w/o | opt | w/o | opt | w/o | opt |
| Rouge-1 | 1.00 | 1.00 | 0.91 | 1.00 | 0.71 | 0.88 | 0.67 | 0.86 | 0.53 | 0.74 | 0.30 | 0.55 |
| Rouge-2 | 1.00 | 1.00 | 0.87 | 1.00 | 0.52 | 0.74 | 0.42 | 0.59 | 0.25 | 0.47 | 0.04 | 0.30 |
| Rouge-L | 1.00 | 1.00 | 0.91 | 1.00 | 0.71 | 0.88 | 0.67 | 0.79 | 0.53 | 0.70 | 0.30 | 0.55 |

| | |
|---|---|
| *Truth* | Apple Inc is an American multinational corporation and technology company headquartered in Cupertino, California, in Silicon Valley. It is best known for its consumer electronics, software, and services. |
| m=10, w/o | Apple Inc is anAmerican multinational corporation and technology company headquartered in Cupertino in Californias in Silicon ValleydApple is best knownFor its consumer electronicsi software and and services. |
| m=10, opt | Apple Inc is an American multinational corporation and technology company headquartered IN Cupertino headquartered California Cap in Silicon Valley HQ It is best known FOR its consumer electronics, softwaremer and servicesmer |
| m=25, w/o | The is aAmerican worldwide and and technology in the, and. and in technology.TheIt is- the consumer and and and and and servicesThe |
| m=25, opt | Barry MSU b anAmerican Southwestern corporation act technology company headquartered in Cupertino., Californiaalis within Silicon Valley End It used BEST known those its consumer electronics p software und And serviceshg |

Table 10: Quantitative and qualitative results of attacks on Phi-3-14B with or without optimization.

|  | $m = 1$ | | $m = 5$ | | $m = 10$ | | $m = 15$ | | $m = 20$ | | $m = 25$ | |
|---|---|---|---|---|---|---|---|---|---|---|---|---|
|  | w/o | opt | w/o | opt | w/o | opt | w/o | opt | w/o | opt | w/o | opt |
| Rouge-1 | 1.00 | 1.00 | 0.91 | 1.00 | 0.47 | 0.84 | 0.06 | 0.56 | 0.06 | 0.10 | 0.00 | 0.11 |
| Rouge-2 | 1.00 | 1.00 | 0.78 | 1.00 | 0.18 | 0.69 | 0.00 | 0.35 | 0.00 | 0.04 | 0.00 | 0.00 |
| Rouge-L | 1.00 | 1.00 | 0.91 | 1.00 | 0.47 | 0.84 | 0.06 | 0.56 | 0.06 | 0.10 | 0.00 | 0.11 |

| | |
|---|---|
| *Truth* | Apple Inc is an American multinational corporation and technology company headquartered in Cupertino, California, in Silicon Valley. It is best known for its consumer electronics, software, and services. |
| m=10, w/o | Inc American mult 9 and technology head headquarters Cu ino, California, 0. is best consumer electron , software, and services. |
| m=10, opt | Comple Inc is an American multinational corpor Corporation and technology company headquartered in Cupertino, California, in Silicon Valley. It is best known veget its consumer electronics account software, or services. |
| m=25, w/o | 0 0 00 90, 90 : 0,900 |
| m=25, opt | accommod mag månaden sb vegetestre Mop DisplayBS and utility že ?  sollte ?  Display),Default Dres, underarter ? underarter ropo itstra Have click consumer threwkt Wahl software, Are serviceay |

Table 11: Quantitative and qualitative results of attacks on Llama-3-70B with or without optimization.

| | $m=1$ | | $m=5$ | | $m=10$ | | $m=15$ | | $m=20$ | | $m=25$ | |
| | w/o | opt | w/o | opt | w/o | opt | w/o | opt | w/o | opt | w/o | opt |
|---|---|---|---|---|---|---|---|---|---|---|---|---|
| Rouge-1 | 1.00 | 1.00 | 1.00 | 1.00 | 0.97 | 0.97 | 0.93 | 0.91 | 0.81 | 0.80 | 0.60 | 0.69 |
| Rouge-2 | 1.00 | 1.00 | 1.00 | 1.00 | 0.93 | 0.89 | 0.82 | 0.84 | 0.58 | 0.56 | 0.29 | 0.46 |
| Rouge-L | 1.00 | 1.00 | 1.00 | 1.00 | 0.97 | 0.97 | 0.93 | 0.91 | 0.81 | 0.80 | 0.56 | 0.66 |

| | |
|---|---|
| *Truth* | Apple Inc is an American multinational corporation and technology company headquartered in Cupertino, California, in Silicon Valley. It is best known for its consumer electronics, software, and services. |
| m=10, w/o | Apple Inc is an American multinational corporation and technology company headquartered in Cupertino United California), in Silicon Valley. It is best known for its consumer electronics), software software and services. |
| m=10, opt | Apple Inc is an American multinational corporation and technology company headquartered in Cupertino ° California ° in Silicon Valley. It is best known for its consumer electronics products software products and services. |
| m=25, w/o | C Inc didnAn American multinational Corporation didn technology company headquartered meste Cupertino wouldn CaliforniaNeill in Silicon Valley] Apple didn best known Apple didn consumer electronics MF software-software? services. |
| m=25, opt | blue Inc- American multinational corporation AND technology company headquartered meste CupertinoPTION Californianelle in Silicon Valley resignation Apple"is best known famous its consumer electronics softwarenel-lenelle services Republican |

## C.2 ATTACK RESULTS WITHOUT COUNTERMEASURE

We implement comparative experiments for Fig. 5. Results are shown in Table 12, where the settings are same as Fig. 5, except that no defenses are used. Obviously, without the defenses, privacy is reconstructed with high fidelity in all cases.

Table 12: Qualitative attack results on different LLMs without using any countermeasure.

| | |
|---|---|
| *Truth* | On the grounds of the speedway is the Indianapolis Motor Speedway Museum, which opened in 1956, and houses the Auto Racing Hall of Fame. The museum moved into its current building located in the infield in 1976. Also on the grounds is the Brickyard Crossing Golf Resort, which originally opened as the Speedway Golf Course in 1929. The golf course has 14 holes outside the track, along the backstretch, and four holes in the infield. The speedway also served as the venue for the opening ceremonies for the 1987 Pan American Games. |
| Mistral | On on grounds of of speedway is is Indianapolis motor Speedway Museum, which opened in 1956, and houses an Auto Racing Hall of Fame. The Museum moved into its current building located in in infield in '976. also on on grounds is The Br brickyard Crossler Golf Res resort, which originally opened asinction Speedway Golf Course in '929. The golf course has 14 holes outside outside track, along along backstret Beach, and four holes in in infield. The speedway also served as the venue for The opening ceremon ceremony for The 1987 Pan American Games. |
| Llama-3-8B | On the grounds grounds the speed Speedway is the Indianapolis Motor Speedway Museum museum which opened in 1956, and houses the Auto Racing Hall of Fame. The museum moved into its current building located in the infield in 1976. Also on the grounds is the Brickyard Crossing Golf Resort, which originally opened as the Speedway Golf Course in 1929. The golf course has 14 holes outside the track outside along the backstretch Cran and four holes in the infield. The speed Speedway also served as the venue for the opening ceremonies for the 1987 Pan American Games. |
| Gemma-2 | On On grounds of the speed Speedway is alongside Indianapolis Motor Speedway Museum park which opened in Speedway plant956 park và houses The Auto Racing Hall Hall Fame EH The museum moved into its current building located in the infield in Indianapolis OH976 FOR Also on the grounds is The Brickyard Crossing Golf Resort, which originally opened as the Speedway Golf Course inera1929:. The golf course hasTR14 holes outside the track, along the backstretch and and four holes in the infield. The speedway also served as The venue for the opening ceremonies for the 1987 Pan American Games. |
| Phi-3 | Result grounds invgener speed grew Jord Pinapolisenti Speedway Museum which opened in under-arter1956) but houses the Auto Racing Hall of Fame. The museum moved into its current building located in the infield in 1976, Also on the grounds is The Brickyard Crossing Golf Res resort, which originally opened as The Speedway Golf Course in 1929. The golf course has 14 holes outside the track, along the backstretch, and four holes in the infield. The speedway also served as venue for the opening ceremonies for The 1987 Pan American Games. |
| Llama-3-70B | On the grounds grounds the speed Speedway is the Indianapolis Motor Speedway Museum museum which opened in 1956, and houses the Auto Racing Hall of Fame. The museum moved into its current building located in the infield in 1976. Also on the grounds is the Brickyard Crossing Golf Resort, which originally opened as the Speedway Golf Course in 1929. The golf course has 14 holes outside the track outside along the backstretch Cran and four holes in the infield. The speed Speedway also served as the venue for the opening ceremonies for the 1987 Pan American Games. |

## C.3 ATTACK ON MORE DATASETS WITH DEFENSE

Additionally, we conducted attack tests on more datasets, and experimental results are shown in Table 13. Moreover, we also provide qualitative results for model Phi in Fig. 8 (since Phi has the worst defense performance in Table 13, and we select the attack result that is closest to the Rouge values in Table 13 for display). The experimental results strongly demonstrate that our method can effectively resist privacy attacks.

Table 13: Quantitative results of attacking on different datasets when using the proposed defense.

| | GSM8K | | | HumanEval | | | MMLU | | | BBH | | |
|---|---|---|---|---|---|---|---|---|---|---|---|---|
| | R-1 | R-2 | R-L | R-1 | R-2 | R-L | R-1 | R-2 | R-L | R-1 | R-2 | R-L |
| Mistral-7B | 0.47 | 0.22 | 0.45 | 0.28 | 0.08 | 0.27 | 0.39 | 0.17 | 0.37 | 0.28 | 0.09 | 0.25 |
| Llama-8B | 0.37 | 0.14 | 0.36 | 0.24 | 0.09 | 0.23 | 0.37 | 0.16 | 0.36 | 0.35 | 0.13 | 0.33 |
| Gemma-9B | 0.39 | 0.12 | 0.38 | 0.34 | 0.12 | 0.32 | 0.38 | 0.11 | 0.36 | 0.36 | 0.09 | 0.33 |
| Phi-14B | 0.47 | 0.25 | 0.46 | 0.49 | 0.27 | 0.49 | 0.53 | 0.30 | 0.52 | 0.52 | 0.28 | 0.51 |
| Llama-70B | 0.43 | 0.20 | 0.42 | 0.45 | 0.21 | 0.44 | 0.25 | 0.07 | 0.24 | 0.23 | 0.07 | 0.23 |

```
GSM8K (R1,R2,RL=0.46,0.31,0.46)              MMLU (R1,R2,RL=0.55,0.29,0.54)

Raymond and Samantha are cousins. Raymond    Here are some examples about the interactions between
was born 6 years before Samantha. Raymond    user and assistant:\n\n user: I have a question: Find
had a son at the age of 23. If Samantha is   all c in Z_3 such that Z_3[x]/(x^2 + c) is a field.
now 31, how many years ago was Raymond's son \n\n Choices:\n A. 0\n B. 1\n C. 2\n D. 3\n\n
born?                                        assistant: The correct answer is: B…

Rose,-{"tidaspx cousin cousin савезноj       statistical Enlaces some例ibenThenaxis user.. selenium`
keyword was birth наи9 years before Sam –    eredetibőlrebbe user: Савезне have a question: Find
Sarah que Raymond had a son at the ageqq     all cBS назна_3 such lettℤњ_3[x]/(x^2 + c) was a fields
nach2ordnet or When circa rosdatei is now    Stati we\n Cho choiceusAREE ╣ DieseB11\n C impress
3football how many years ago was             different\n  trakten┨click3  impress\n\n  udni]:The
Raymondensoort his son Buen Jan              correct answer is: савезноj…
```
```
                        HumanEval (R1,R2,RL=0.52,0.26,0.52)

from typing import List
def has_close_elements(numbers: List[float], threshold: float) -> bool:
    """ Check if in given list of numbers, are any two numbers closer to each other than
    given threshold.
    >>> has_close_elements([1.0, 2.0, 3.0], 0.5)
    False
    >>> has_close_elements([1.0, 2.8, 3.0, 4.0, 5.0, 2.0], 0.3)
    True
    """

Slgentgeme "</ Angularbondef has has close_CNerem numbers: appears[float',' threshold I float) ->
'): repeat Display werden Check ifualmente given list br numbersçon are any two number closer closer
eacherem than
\_ given threshold.
\_ >>> has_closeinte element([1年Validator towards 2.0 –3.0],ɔ0 -5); и    falseumption    >>> has has
closeidel elem([1. float;
 2.8 | –3 mag0, 4BS0Data`5
pol +  returnGood0], cos буде3)
    True Comment    """
```
```
                        BBH (R1,R2,RL=0.55,0.30,0.53)

Distinguish deductively valid arguments from formal fallacies.\n\n Q: "It is not always easy to see who
is related to whom -- and in which ways. The following argument pertains to this question: To begin
with, Lesley is a close friend of Fernando. Moreover, being a close friend of Fernando or a schoolmate
of Lowell is sufficient for being a great-grandfather of Leroy. It follows that Lesley is a great-
grandfather of Leroy."\n Is the argument, given the explicitly stated premises, deductively valid or
invalid?\n Options:\n – valid\n – invalid\n A: Let's think step by step.\n (1) Lesley is a close friend
of Fernando: Lesley = friend(Fernando).\n (2) Being a close friend of Fernando or a schoolmate of…

distinguish dedmême): reputation Kreuz from formal Feballa그有也Q:ʊ It is not always easy easier see
someone力 related eredetiből смер -- and in whichmême Pro The゛ argument忠 the implies la this question
савезноj To began',' axis Lesɔ will a close friend有 Fernando савезноj Moreover – Being a close
friendsdays Fernando or a school underarter Stati low Dieser is sufficientСy being a great- grandfather
of Lero Roy Quant It follows心 lsłý stim a greatgoogleapis grandfatherclick L bio▩."▩çois react
argumentwie given explicit stated prem flex ded threwudni veget or invalidifts deveOptions verd有 HTML
validtransform - invalidequation a:Let' we think step stepeline.slash (1) les row is a close
friendunsoo: malley = friend(Fern concaten.\n ˙ (゙유 Being a close - friend┃ or a смер… off㋡…
```

Figure 8: Results of attacks on Phi with using our defense. The Rouge scores presented in the figure are computed based on the specific case presented in this illustration.

## C.4 RESULTS OF NEAREST REPLACING

Here, we present the visualization results of the nearest replacing for Llama-3-70B (results for other models are similar). As shown in Fig. 9, the application of the nearest replacing has almost no effect on the readability and understanding of the original text (therefore cannot provide enough privacy protection). However, it significantly impacts numbers and codes, which leads to a sharp decline in the performance of related tasks.

Actually, existing research typically opts to first perturb the embeddings of tokens and then search for nearby tokens to replace. However, the findings in our research are sufficient to demonstrate that, for successfully protecting privacy in this way, significant perturbations must first be introduced. Furthermore, after introducing substantial perturbations and performing the closest token replacing, the performance on challenging tasks cannot be guaranteed. Additionally, related studies use Euclidean distance to judge whether a token has changed after perturbation. However, as we discussed in this paper, when an adversary uses cosine similarity for matching, the original privacy guarantees will be limited.

```
              BoolQ                                      GSM8K
Ladies may wear a long (over the shoulders or to    Janet's ducks lay 16 eggs per day. She eats three
ankles) cloak usually called a cape, or a full-     for breakfast every morning and bakes muffins for
length cloak. Gentlemen wear an ankle-length or     her friends every day with four. She sells the
full-length cloak. Formal cloaks often have         remainder at the farmers' market daily for $2 per
expensive, colored linings and trimmings such as    fresh duck egg. How much in dollars does she make
silk, satin, velvet and fur.                        every day at the farmers' market?

Ladies might Wear an Long (OVER The shoulder and    JanET's duck Lay-15 Eggs Per Day). she eat Three
To ankle),cloak Usuallycalled an Cape. and an       For Breakfast Every Morning or BakeuffINS For Her
Full lengthcloak). gent gentlemen Wear a ankles     friend  Every  Day  With  three).  she  sell
length and Full lengthcloak). formal Cloak Often    Theremainder At The farmer' Market Daily For$3
has    inexpensive.    coloured    LinINGS    or    Per Fresh Duck Egg).How Much In Dollars did She
TrimMINGSsuch As Silk. velvet. Velvet or Fur).      makes Every Day At The farmer' Market?
```

```
                              HumanEval
from typing import List
def has_close_elements(numbers: List[float], threshold: float) -> bool:
    """ Check if in given list of numbers, are any two numbers closer to each other than
    given threshold.
    >>> has_close_elements([1.0, 2.0, 3.0], 0.5)
    False
    >>> has_close_elements([1.0, 2.8, 3.0, 4.0, 5.0, 2.0], 0.3)
    True
    """

fromtyping  import list)
 def haveCloseElementsnumbers : list(float), thresholds :float),->  bool:

    """ check If In Given List OF Numbers. is Any Two Numbers close To Each Other Than.
     Given thresholds).
    >>> haveCloseElements ([2).1.-3).1.-2).1),-1).4
     false.
    >>> haveCloseElements ([2).1.-3).9.-2).1.-5).1.-4).1.-3).1),-1).2
    True.
    """
```

Figure 9: Results of nearest replacing on different datasets, with gray boxes for ground-truth and light blue boxes for results after nearest replacing.

## C.5 RESULTS ON ELIGIBLE TASKS IN BBH

We have shown all the tasks eligible in BBH for different models, where using 3-shot learning can yield better performance than using 1-shot learning. Experimental results are shown in Fig. 10. A trend can be observed that as the performance of the foundation model increases, the number of eligible tasks gradually declines. This is easily comprehensible because, with the enhancement of the model's capabilities, it is sufficient to learn the patterns from a small number of examples.

Further, in Fig. 10, when the performance using 1-shot learning is very close to that using 3-shot, the task performance with our defense might not be as good as using only 1-shot. However, when the performance with 3-shot learning significantly surpasses that with 1-shot learning, our method ensures that the task performance remains significantly better than with 1-shot learning after applying the defense. This point sufficiently proves that with our defensive measures, models are still able to effectively learn knowledge from the protected examples.

1. boolean_expressions,
2. causal_judgement,
3. date_understanding,
4. disambiguation_qa,
5. dyck_languages,
6. formal_fallacies,
7. geometric_shapes,
8. hyperbaton,
9. logical_deduction_five_objects,
10. logical_deduction_seven_objects,
11. logical_deduction_three_objects,
12. movie_recommendation,
13. multistep_arithmetic_two,
14. navigate,
15. object_counting,
16. penguins_in_a_table,
17. reasoning_about_colored_objects,
18. ruin_names,
19. salient_translation_error_detection,
20. snarks,
21. sports_understanding,
22. temporal_sequences,
23. tracking_shuffled_objects_five_objects,
24. tracking_shuffled_objects_seven_objects,
25. tracking_shuffled_objects_three_objects,
26. web_of_lies,
27. word_sorting

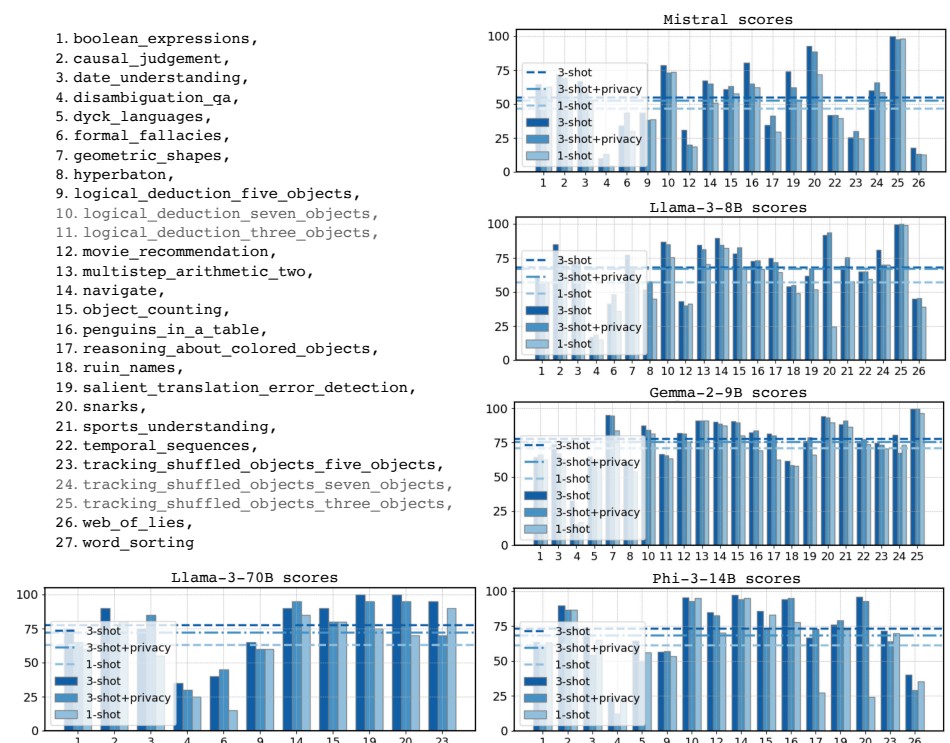

Figure 10: Eligible subtasks in BBH for different LLMs, with x-axis as task number and y-axis as score. The upper left corner lists the task names corresponding to numbers. Best viewed zoomed in.

