# OpenReview forum: "Safeguard User Privacy in LLM Cloud Services"
_ICLR.cc/2025/Conference — ICLR 2025 Conference Withdrawn Submission_

### Official Review · Reviewer_SfaZ · 2024-10-28

**Soundness:** 2
**Presentation:** 3
**Contribution:** 3
**Rating:** 6
**Confidence:** 2

**Summary:**

This paper provides a detailed analysis of the embedding vector space in large language models (LLMs) and examines the privacy vulnerabilities associated with reconstruction attacks.

-- The authors show that the embedding space of tokens is sparse by computing Inclusion Ratio, and they demonstrate that shallow layers do not significantly alter the direction of embedding vectors by calculating the Cosine distance. As a result, privacy is highly susceptible to adversaries.

-- The paper proposes a distributed inference paradigm designed to resist reconstruction attacks on LLMs and evaluates its effectiveness.

-- The authors also discuss the computational overhead of their proposed method and investigate the impact of techniques such as quantization, perturbation, and replacement.

**Strengths:**

1. The paper presents detailed and comprehensive experiments and results to support its two main conclusions about the embedding space in large language models (LLMs).

2. The paper is well-written and easy to follow. Thank you for the effort.

**Weaknesses:**

The paper does not provide essential background information on large language models (LLMs) and reconstruction attacks in LLMs, making it challenging for researchers outside this specific field to understand.

I also have some questions about the reconstruction attack in LLMs:

1. What information does the attacker use to reconstruct the user input? Is it based solely on the hidden states?
2. What approach did you take in the evaluation section? Are you using the same attack methodology as described in prior work?

**Questions:**

Thank you for your efforts on the paper. I have a few questions regarding the proposed protection for both the input and output of the LLM:

1. What does the output look like? Is it in plain text, or is it noise-perturbed?
2. Is it feasible to apply the same protection method to the LLM's output as was applied to the input?
3. How might the reconstruction attack interact with the LLM's output? Specifically, could the output potentially aid in reconstructing the original input?

---

### Official Review · Reviewer_TpTn · 2024-11-01

**Soundness:** 1
**Presentation:** 3
**Contribution:** 2
**Rating:** 3
**Confidence:** 4

**Summary:**

The paper considers the privacy-preserving usage of LLMs in cloud scenarios.
The authors first observed that the token embeddings of modern LLMs typically are sparse, and make another claim that the LLMs typically 'work' (i.e., the LLM's transformer blocks process embeddings and subsequent hidden states) in a subspace orthogonal to the embedding space.

Based on these observations, they propose a privacy defense that aims at protecting sensitive information in the users' prompts that are sent to the LLM in the cloud: First, the users locally run the LLM to compute the hidden states of their prompt, and send the hidden states instead of the prompt to the cloud LLM for completion (generation of the response). Second, in the local, layer-wise computations of the hidden states, random contractions are applied to prevent attackers from inverting and recovering the original embeddings and hence from reconstructing the original tokens.

The authors derive suitable parameters for their method and perform an extensive experimental evaluation of their method using various tasks and datasets. The results appear promising in the sense that the method is effective against server-side adversaries who can use embedding inversion attacks and only has small impact on most tasks' utility.

**Strengths:**

I like the idea of considering the effect of an LLM's layers on token reconstruction (embedding inversion) attacks.

The experimental evaluation of the defense method is comprehensive across a variety of tasks and indicates that the method seems to work -- at least against the (non-adaptive) attacks described in the paper.

**Weaknesses:**

Modern LLMs have hidden/embedding dimensions that are in the thousands. A known effect in high-dimensional vector spaces is that random vectors are expected to be nearly orthogonal.
Since you are empirically computing angles between residual vectors ("cumulative sums J_1 + J_2") and *random* token embedding vectors, I am wondering if all you measured in Proposition 1 (and in A.2) was just an observation of this phenomenon of high-dimensional vectors spaces.
In fact, I do not see why observed (but expected) orthogonality with random embedding vectors would support your claim that embeddings and the residuals from the LLM's layers are actually in orthogonal (proper) subspaces: Both may well take place in exactly the same, i.e., improper, (sub)space.
I'd suggest considering a more suitable experiment to support (or refute) the claim of Prop. 1:
One idea that comes to my mind is computing the PCA of embedding vectors and determining if there is a (proper!) subspace spanned by the most significant (say, up to some threshold) principal components.
Then you could check if and to what extent the residual terms from the LLM's layers are orthogonal to this principal subspace. (Disclaimer: I make no claim that this is the best method :-).)

Moreover, as defense approach you propose to randomly contract the hidden states h^(i-1) as to make the proportion of the residual terms larger and thus increase the "nonlinearity".
However, as you just claimed in Prop. 1, these residuals/nonlinearities are supposedly near-orthogonal to the embeddings, so they should only have a marginal effect on the angles to the token embeddings -- but these angles are exactly what the attacker computes to find the nearest candidate tokens!
More mathematically, if we write u=h^(i-1) for the unmodified hidden state, v=J_1+j_2 for the residual/nonlinear term that supposedly is orthogonal to the token embeddings, w to be a candidate token embedding, and x=1/p a random scaling factor, we get the dot product <v,w> \approx 0 due to the (claimed) near-orthogonality, and hence <xu + v, w> = x<u,w> + <v,w> \approx x<u,w> .
And as you stated: "cosine distance is insensitive to the magnitude", so shrinking u=h^(i-1) and hence the <u,w> by a factor x won't change these angles!
So in fact, your Prop. 1 implies that the proposed defense should NOT be very effective -- or conversely, if the defense is effective, Prop. 1 may be wrong, i.e., the embeddings and residuals are in the same (sub)space.

While it's nice to know that your methods is compatible with quantization, I am not sure if/how relevant it is for the theme of the paper (mix of concepts: privacy vs. size/efficiency).

Lastly, while your experiments indicate that the method is effective against the described reconstruction attacks, I want to remark that empirical privacy evaluations without formal privacy guarantees should be taken with a grain of salt (cf. Aerni et al., "Evaluations of Machine Learning Privacy Defenses are Misleading", CCS 2024). Therefore, I remain skeptical about the long-term validity and significance of the method's privacy protection capability.
If the LLM can still accurately answer queries based on modified hidden states, it seems unintuitive that the sequence of tokens in the query cannot be better recovered (provided that the response has non-trivial dependencies on the query, e.g., for text summarization tasks, or when the reply is more complex than a simple 'yes'/'no', a number, or similar). Think: What would an LLM with your defense respond to prompts like "<Some text here>. Please quote the preceding text in verbatim."? Therefore, I wonder if more sophisticated attacks that are adapted to the defense approach (e.g., train a model that learns from collected 'modified hidden state' -> 'actual token/embedding' pairs) could be more successful.

**Questions:**

L91,92: As far as I could see in their papers, (Yue 2022) perturb gradients of weights using DP-SGD and (Liu 2024) perturb weights directly with zeroth-order optimization; neither paper perturbs embeddings.

L105, L447: Do you mean "private" instead of "privacy"? (What does it mean to "treat the examples [...] as privacy"?)

L188: What does it mean for a vector to be "almost included" in some set?

L215: Check reference to Fig. 1?

L365-366: "with the conditions of Rouge-1 < 0.5, Rouge-2 < 0.3, Rouge-L < 0.5, it is sufficient for the reconstruction to compromise a significant amount of privacy information from the original data" <- I am unsure what you want to say. So small reconstruction (Rogue) scores are good for privacy leakage? Please rephrase/express more clearly.

L539: (You already refer to this as a limitation in L539, but I want to ask explicitly:) If the hidden states have to be computed locally, and only the completion/generation of the response is delegated to the server, what is the point of doing so? (Especially in the 'choice' tasks where the response is typically very short, I don't see much benefit in doing so. Would the overhead for computing the response locally not be almost negligible?)

---

### Official Review · Reviewer_RMyC · 2024-11-03

**Soundness:** 3
**Presentation:** 2
**Contribution:** 2
**Rating:** 3
**Confidence:** 4

**Summary:**

This paper finds the sparse features of large language models, which leads to vulnerability of distributed inference. The paper enhances nonlinear modules against reconstruction attacks. Experimental results validates the effectiveness of the proposed method.

**Strengths:**

1. The authors propose a new solution to the privacy issues of large language models.
2. The intuition behind the defense is straightforward.
3. The proposed method is effective even for utilized in solving math problems.

**Weaknesses:**

1.  Previous works [1,2] have highlighted the sparse nature of large language models. The statement of first contribution appears overstated and needs more precise positioning of its novelty.
2. The methodology primarily relies on empirical reconstruction attacks against watermark defenses (line 207). While the subsequent statistical tests are rigorously designed, this approach may be limited compared to more sophisticated attack strategies proposed in [3,4] that leverage optimal strategies based on prior knowledge. The reliability of these empirically-based assumptions remains questionable against advanced adversarial approaches.
3. The experimental section lacks implementation of important baseline methods [5,6]. This omission makes it difficult to properly evaluate the effectiveness of the proposed approach in the context of existing solutions. Including these comparisons would provide a more comprehensive evaluation framework.

* [1] Zhang Z, Xiao C, Qin Q, et al. Exploring the Benefit of Activation Sparsity in Pre-training[J]. ICML, 2024.
* [2] Liu Z, Wang J, Dao T, et al. Deja vu: Contextual sparsity for efficient llms at inference time[C]//International Conference on Machine Learning. PMLR, 2023: 22137-22176.
* [3] Huang Y, Gupta S, Song Z, et al. Evaluating gradient inversion attacks and defenses in federated learning[J]. Advances in neural information processing systems, 2021, 34: 7232-7241.
* [4] Tong M, Chen K, Yuang X, et al. On the Vulnerability of Text Sanitization[J]. arXiv preprint arXiv:2410.17052, 2024.
* [5] Mai P, Yan R, Huang Z, et al. Split-and-Denoise: Protect large language model inference with local differential privacy[J]. arXiv preprint arXiv:2310.09130, 2023.
* [6] Pang Q, Zhu J, Möllering H, et al. Bolt: Privacy-preserving, accurate and efficient inference for transformers[C]//2024 IEEE Symposium on Security and Privacy (SP). IEEE, 2024: 4753-4771.

**Questions:**

1. Given the introduction of additional privacy protection mechanisms, is further measurement of inference time and storage overhead necessary? Does this privacy protection scheme incur additional costs?
2. The evaluation metrics for utility should be explained in more detail. Are these metrics consistent across different datasets?

---

### Official Review · Reviewer_vrh6 · 2024-11-03

**Soundness:** 3
**Presentation:** 3
**Contribution:** 3
**Rating:** 6
**Confidence:** 3

**Summary:**

This paper investigates the privacy leakage issues associated with LLMs. The authors identify two key findings: the token embedding space is highly sparse, and LLMs primarily operate in orthogonal subspaces. These factors make the privacy of LLMs vulnerable to leakage. To address this, the authors propose a distributed privacy protection strategy to effectively defend against privacy attacks. Experimental results demonstrate that the proposed mechanism achieves a good balance between privacy, practicality, and runtime memory efficiency.

**Strengths:**

The structure of this paper is logical; The problem being addressed is very important for the healthy development of the LLM-related field; The authors obtained some interesting phenomena through preliminary experiments and proposed innovative solutions; The experiments conducted in this paper are thorough, and the results are convincing.

**Weaknesses:**

Some key design statements are unclear and require further explanation; additionally, the rationale behind certain proposals needs to be explained in more detail.

**Questions:**

1. The sparsity of the token embedding space is significant and is closely related to the design of LLMs. However, this issue is not difficult for LLM designers or providers to address. The authors should clarify why the designers have not implemented upgrades to better illustrate the motivation behind this paper.
2. In section 3.2.2, the authors only compare and consider the performances of Euclidean distance and cosine similarity. However, there are many other similarity measures available. The authors need to provide further explanation for why they only considered these two methods.
3. What types of companies represent malicious service providers in the attack model in the real world, and what are their objectives for carrying out such attacks? What benefits can these privacy leaks bring them?
4. The currently proposed mechanism has high computational requirements, which may further limit the range of users. How do the authors plan to address the impacts and limitations posed by these computational demands?

---

### Official Review · Reviewer_bLyt · 2024-11-03

**Soundness:** 2
**Presentation:** 4
**Contribution:** 3
**Rating:** 5
**Confidence:** 4

**Summary:**

The paper demonstrates the privacy vulnerabilities in LLM in terms of sparsity of embedding space and privacy breaches from directions. Then they propose a privacy-enhancing inference method by randomly scaling the output of first i layers. The empirical evaluation validates the balance between privacy, utility, and runtime memory efffciency.

**Strengths:**

- The paper is well-written, with clear description of motivation and empirical evaluation.
- The proposed method has little impact on the utility.

**Weaknesses:**

**Privacy concerns**

The paper argues that DP could leak to privacy leakage and doesn't adopt the DP guarantee. However, there is no empirical comparison between the protection levels for DP-based method and their proposed method [1][2]. Furthermore, they only use the reconstruction attack to evaluate the protection level of their method. However, from the reconstruted results, we can see that certain attributes/key words in the context can still be inferred by the attacker. More attacks, such as attribute inference attack, are desired.

From Figure 1, only the context is protected in the framework, while the instruction could also leak user's private information. Another inherent limitation of this framework is that if the LLM could return highly accurate response, then it's inevitable that the response would reveal sensitive information of the user. It's suggested that the authors could apply certain post-processing operations on the response to address the dilemma [3].

**Attack methods**

The paper considers a optimization-based attack method. In reality, the attackers could use a variety of approaches to reconstruct the original token. For example, they can train a more transformer-based model to infer the input from the privatized hidden representation. The authors may conduct experiments on more advanced attacks.

[1] Feyisetan, O., Balle, B., Drake, T., & Diethe, T. (2020, January). Privacy-and utility-preserving textual analysis via calibrated multivariate perturbations. In Proceedings of the 13th international conference on web search and data mining (pp. 178-186).

[2] Mattern, J., Weggenmann, B., & Kerschbaum, F. (2022, July). The Limits of Word Level Differential Privacy. In Findings of the Association for Computational Linguistics: NAACL 2022 (pp. 867-881).

[3] Chen, Y., Li, T., Liu, H., & Yu, Y. (2023). Hide and seek (has): A lightweight framework for prompt privacy protection. arXiv preprint arXiv:2309.03057.

**Questions:**

- What's the performance of the method if both the context and instruction are protected?

- How to decide the number of layers to be deployed on the user device?

---

### Official Review · Reviewer_z2sV · 2024-11-04

**Soundness:** 3
**Presentation:** 3
**Contribution:** 3
**Rating:** 6
**Confidence:** 2

**Summary:**

This paper conducts a comprehensive empirical study of privacy vulnerabilities in LLMs,  and design a distributed privacy-preserving inference paradigm which can effectively resist privacy attacks.

**Strengths:**

1. This paper is well-organized, with clear motivations and analysis of privacy vulnerabilities in LLMs.
2. The experimental results are persuasive and promising.

**Weaknesses:**

1. I am not sure with the novelty of the technical contribution in this paper, is it only adjusting the two functions so that the nonlinear modules are amplified ?
2. How to shrink the hidden state ? Can you elaborate it more clearly?
3. Did you consider other attacks beyond the reconstruction attack ?

**Questions:**

See weaknesses

---

### Note · Authors · 2024-11-15

**Comment:**

Some related work has only been evaluated on simple tasks and clearly cannot be applied to difficult ones. For instance, adding random noise to prompt words until they become unrecognizable and then transmitting them to the LLM in cloud, which, after returning the results, uses an additional local model for denoising. Can this approach really work for fine-grained tasks? Those papers did not conduct such experiments, and I also feel there's no need to compare with them unless the locally trained denoising model can achieve representation capabilities comparable to the LLMs (is this achievable for users?). Otherwise, I believe it's clearly not feasible. In line with the reviewer's comments and requirements (ensuring complete privacy for both input and output, without affecting the usability of the large model, while also considering practicality), I have decided to focus my future research on trusted execution environments. Perhaps extending TEE to AI accelerators is the only viable solution.

**Withdrawal Confirmation:**

I have read and agree with the venue's withdrawal policy on behalf of myself and my co-authors.